# Competition between Tropomyosin, Fimbrin, and ADF/Cofilin drives their sorting to distinct actin filament networks

Jenna R Christensen[1], Glen M Hocky[2,3], Kaitlin E Homa[1], Alisha N Morganthaler[1], Sarah E Hitchcock-DeGregori[4], Gregory A Voth[2,3,5,6], David R Kovar[1,7]*

[1]Department of Molecular Genetics and Cell Biology, The University of Chicago, Chicago, United States; [2]Department of Chemistry, The University of Chicago, Chicago, United States; [3]James Franck Institute, The University of Chicago, Chicago, United States; [4]Department of Pathology and Laboratory Medicine, Robert Wood Johnson Medical School, Rutgers University, New Brunswick, United States; [5]Computation Institute, The University of Chicago, Chicago, United States; [6]Institute for Biophysical Dynamics, The University of Chicago, Chicago, United States; [7]Department of Biochemistry and Molecular Biology, The University of Chicago, Chicago, United States

**Abstract** The fission yeast actin cytoskeleton is an ideal, simplified system to investigate fundamental mechanisms behind cellular self-organization. By focusing on the stabilizing protein tropomyosin Cdc8, bundling protein fimbrin Fim1, and severing protein coffin Adf1, we examined how their pairwise and collective interactions with actin filaments regulate their activity and segregation to functionally diverse F-actin networks. Utilizing multi-color TIRF microscopy of in vitro reconstituted F-actin networks, we observed and characterized two distinct Cdc8 cables loading and spreading cooperatively on individual actin filaments. Furthermore, Cdc8, Fim1, and Adf1 all compete for association with F-actin by different mechanisms, and their cooperative association with actin filaments affects their ability to compete. Finally, competition between Fim1 and Adf1 for F-actin synergizes their activities, promoting rapid displacement of Cdc8 from a dense F-actin network. Our findings reveal that competitive and cooperative interactions between actin binding proteins help define their associations with different F-actin networks.

*For correspondence: drkovar@ uchicago.edu

**Competing interests:** The authors declare that no competing interests exist.

## Introduction

The self-organization of complex structures from interactions between basic components is a general phenomenon of chemistry and material sciences, as well as more complicated biological systems (*Karsenti, 2008*). Examples include the generation of self-segregating PAR domains in the developing *C. elegans* embryo (*Hoege and Hyman, 2013*; *Goldstein and Macara, 2007*) and the organization of a mitotic spindle around DNA-coated microspheres (*Heald et al., 1996*). How individual interactions within the cell coalesce to generate complex patterns or structures remains a fundamental biological question. The actin cytoskeleton is an ideal system to study complex cellular self-organization. Multiple functionally diverse F-actin networks, each with a distinct architecture and dynamics, assemble at the correct time and place within a single crowded cytoplasm. Distinct sets of actin binding proteins (ABPs) help to define the characteristics of each F-actin network by performing tasks such as actin filament (F-actin) nucleation, bundling, severing, and capping (*Pollard, 2016*). Therefore, proper localization of ABPs to the correct network is crucial to generate F-actin networks

**eLife digest** Cells use a protein called actin to provide shape, to generate the forces needed for cells to divide, and for many other essential processes. Inside a cell, individual actin proteins join up to form long filaments. These actin filaments are organized in different ways to make networks that have distinct properties, each tailored for a specific process. For instance, bundles of straight actin filaments help a cell to divide, whereas a network of branched actin filaments allows cells to move.

The different proteins that bind to actin filaments influence how quickly actin filaments are assembled and organized into networks. Therefore, many of the properties of an actin filament network are due to the actin binding proteins that are associated with it. Two actin binding proteins called fimbrin and cofilin associate with a type of actin filament network known as the actin patch. A third actin binding protein called tropomyosin associates with a different network that forms a ring. It is not known how particular actin binding proteins choose to associate with one actin network instead of another.

Christensen et al. used a fluorescence microscopy technique to study how fimbrin, cofilin and tropomyosin associate with different actin networks in a single-celled organism called fission yeast. This technique involved incubating actin and actin binding proteins together in a microscope chamber. The experiments show that some actin binding proteins, like tropomyosin, cooperate to bind to actin. Individual tropomyosin molecules find it difficult to bind actin filaments on their own, but once one tropomyosin molecule is attached to the filament, others rapidly join to coat the filament.

On the other hand, some actin-binding proteins compete for binding to filaments. For example, the binding of fimbrin to actin filaments causes tropomyosin to be removed from the actin network. Further experiments revealed that fimbrin and cofilin work with each other to rapidly generate a dense actin network and displace tropomyosin. Together, the findings of Christensen et al. suggest that competitions between actin binding proteins determine which actin binding proteins are associated with an actin network.

The next challenge is to understand how the most competitive actin-binding proteins are kept off actin networks where they do not belong. Further studies will shed light on how these interactions cause large changes in how the cell is organized.

defined for specific processes. The biochemical activity and cellular functions of many individual ABPs have been well-studied. However, we are only beginning to understand how ABPs function in concert, how they compete with each other for association with individual actin filaments, and how these interactions contribute to the proper sorting of ABPs to diverse F-actin networks on a whole-cell scale (*Michelot and Drubin, 2011*; *Skau and Kovar, 2010*; *Jégou and Romet-Lemonne, 2016*).

Fission yeast is an ideal simplified system in which to study the underlying molecular mechanisms behind F-actin network self-organization (*Kovar et al., 2011*). Fission yeast has three primary actin cytoskeleton networks, in which all of the actin filaments are assembled by a distinct actin assembly factor: endocytic actin patches (Arp2/3 complex), polarizing actin cables (formin For3), and the cyto-kinetic contractile ring (formin Cdc12). Moreover, each of these F-actin networks contains a distinct set of ABPs. We hypothesize that ABP competition for association with actin filaments is critical for their proper sorting to distinct F-actin networks. We previously discovered that competition between ABPs tropomyosin Cdc8, fimbrin Fim1, and ADF/cofilin Adf1 results in the exclusion of tropomyosin from actin patches (*Skau and Kovar, 2010*). Here, we use a combination of multi-color TIRF micros-copy (TIRFM) of reconstituted F-actin networks and mathematical modeling to elucidate the underly-ing molecular mechanisms behind this series of competitive ABP interactions in fission yeast. By understanding ABP competition at a mechanistic level, we can gain insight into how an ABP's intrin-sic physical properties help dictate its interactions with other ABPs and its association with specific F-actin networks. In this study, we have found that different modes of active and passive competition exist between different ABPs. Furthermore, we have determined that cooperativity affects ABP com-petition by defining the ability of ABPs to both associate and be dissociated from an F-actin net-work. Finally, we have found that the combination of cooperative and competitive interactions

between a set of ABPs defines the ABP composition and F-actin organization of the associated F-actin network.

## Results

### Direct visualization of tropomyosin Cdc8 cooperative loading onto actin filaments

Tropomyosin has been implicated in ABP sorting and F-actin network organization in many organisms including fission yeast (*Gunning et al., 2015*; *Skau and Kovar, 2010*; *Clayton et al., 2010*). Tropomyosin is a two-chained, parallel coiled-coil composed of two polypeptide chains with a characteristic heptad-repeat of hydrophobic residues (*Figure 1—figure supplement 1*). Individual tropomyosin coiled coils associate end-to-end to form continuous tropomyosin cables that extend along both sides of the helical actin filament (*Hanson and Lowy, 1963*; *Skoumpla et al., 2007*). Vertebrate tropomyosins span six or seven actin subunits in the filament. Yeast tropomyosins are shorter: the two *S. cerevisiae* tropomyosin isoforms span four or five actins, while *S. pombe* tropomyosin Cdc8 (hereafter called Cdc8) spans four actin subunits. Although Cdc8, like other tropomyosins, has been purified and characterized in steady state bulk assays (*Cranz-Mileva et al., 2015*; *Skau et al., 2009*; *Skoumpla et al., 2007*), our general mechanistic understanding of how tropomyosins load onto actin filaments and are affected by other ABPs is unclear. Multi-color TIRFM has been successfully utilized as a sensitive probe to study the detailed interactions between multiple ABPs and F-actin (*Bombardier et al., 2015*; *Jansen et al., 2015*; *Winkelman et al., 2016*). However, fluorescently labeling tropomyosins is generally problematic as mutations or insertions within the protein can potentially disrupt its coiled-coil structure (*Greenfield and Hitchcock-DeGregori, 1995*). Additionally, tropomyosins labeled on the N- or C-terminus are not fully functional, as the presence of a label blocks end-to-end associations between tropomyosin molecules (*Brooker et al., 2016*). Therefore, we created three distinct Cdc8 mutants, each containing an engineered cysteine mutation that could be labeled for visualization by TIRFM (*Figure 1—figure supplement 1*, Methods). We examined the functionality of each Cdc8 mutant in vitro and in vivo to identify the mutant most similar to wild-type Cdc8. Two mutants, Cdc8(I76C) and Cdc8(D142C), bound F-actin similarly to wild type Cdc8 (*Figure 1—figure supplement 1B–C*), were able to be labeled and visualized by TIRFM (100% of actin filaments associated with tropomyosin, *Figure 1—figure supplement 1D*), and caused only very mild cytokinesis defects as the sole copy of Cdc8 in fission yeast (*Figure 1—figure supplement 2A–D*). Therefore, we chose one of these mutants, Cdc8(I76C), for further study in TIRFM experiments.

We utilized two-color TIRFM to examine the loading characteristics of Cy5-labeled Cdc8 on assembling Alexa 488-labeled actin filaments over a range of concentrations (*Figure 1A–B*, *Video 1*). At concentrations below 1 μM, Cdc8 was not observed to bind F-actin. However, at 1 μM Cdc8, short stretches of Cdc8 were observed to load on actin filaments. These dynamic stretches varied in size over time, but could be as small as the observable limit of ~100 nm and generally remained within the constraints of ~0.5–2 μm (~2% of total actin filament coverage) over time. At 1.25 μM Cdc8, spreading of Cdc8 cables from initial 'seeds' was observed, with ~40% of actin filament sites coated with Cdc8. At concentrations of 2 μM Cdc8 and higher, ~98% of F-actin was coated with Cdc8. This rapid shift in actin filament occupancy over a small Cdc8 concentration range indicates a high degree of cooperativity. One possibility is that Cdc8's high cooperativity is a result of the previously demonstrated end-to-end associations between tropomyosin molecules ('end-to-end cooperativity') (*Figure 1C*) (*Caspar et al., 1969*; *Greenfield et al., 2006*). However, the cooperativity of muscle tropomyosin on F-actin has been found to not directly correlate with its end-to-end binding ability (*Willadsen et al., 1992*), suggesting that other interactions may also influence tropomyosin's cooperativity on F-actin (*Tobacman, 2008*). We therefore considered whether indirect interactions between Cdc8 cables on opposing sides of an actin filament or via long-range interactions along the length of the F-actin lattice ('indirect cooperativity') also contribute to Cdc8's high cooperativity (*Figure 1C*). We focused on investigating a potential role for indirect cooperativity in Cdc8 loading onto an actin filament by observing (*Figure 2*) and quantifying (*Figure 3*) Cdc8 loading events in detail under conditions near the inflection point of Cdc8's cooperativity (1.25 μM Cdc8).

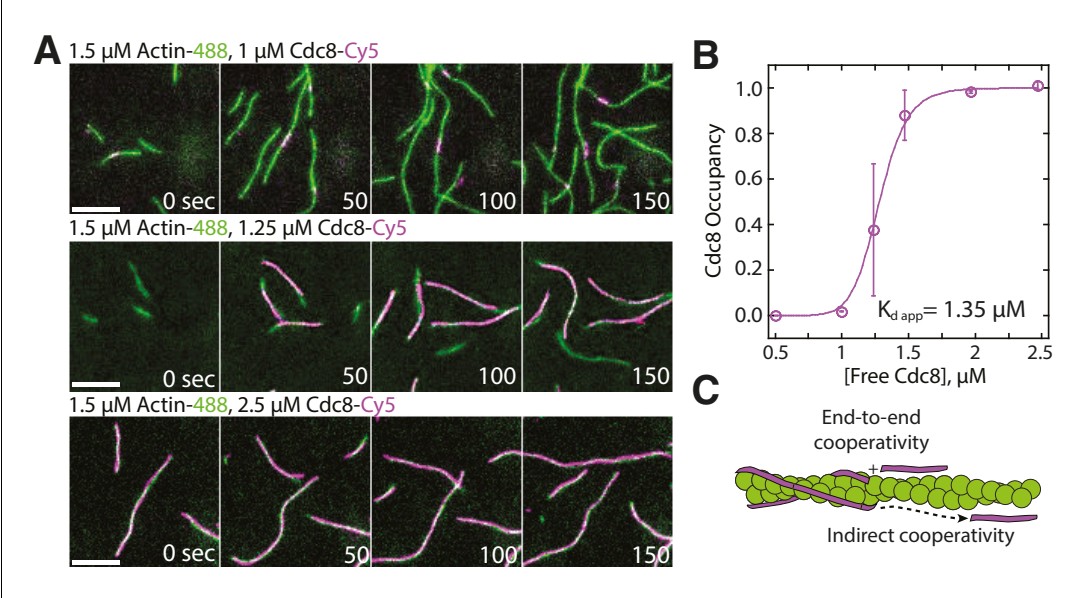

**Figure 1.** Tropomyosin Cdc8 loads cooperatively onto actin filaments. (**A**) Two-color TIRFM of 1.5 µM Mg-ATP actin (15% Alexa 488 labeled) with a range of concentrations of tropomyosin Cdc8 (Cy5-labeled). Scale bar, 5 µm. (**B**) Plot of the fraction of actin filament bound by Cdc8 ('Cdc8 occupancy') over free Cdc8 dimer concentration. Data were fit to a Hill function, revealing a Hill coefficient >1 (Hill=14.6), that indicates cooperativity. Error bars represent standard error of the mean; n = 2 reactions. (**C**) Schematic of Cdc8 loading onto an actin filament. Observed cooperativity of Cdc8 could be the result of end-to-end binding of tropomyosin molecules ('End-to-end cooperativity') and/or indirect interactions between tropomyosin molecules via changes in the actin filament ('Indirect cooperativity').

The following figure supplements are available for figure 1:

**Figure supplement 1.** Characterization of Tropomyosin Cdc8 mutants L38C, I76C, and D142C in vitro.

**Figure supplement 2.** Characterization of Tropomyosin Cdc8 mutants L38C, I76C, and D142C in vivo.

## Visualization of two distinct tropomyosin Cdc8 cables loading onto a single actin filament

Examination of timelapse images (*Figure 2A*) and kymographs (*Figure 2B*) of an elongating actin filament and associated tropomyosin Cdc8 molecules revealed the loading and spreading behavior of Cdc8 on F-actin. Cdc8 loading is complex and characterized by several initial 'seed' events, followed by Cdc8 cable extension along the actin filament toward both the barbed and pointed ends, interrupted by frequent stops and starts (*Figure 2A–B*). Sites of initial Cdc8 seed association were identified by an increase in Cdc8 fluorescence at the site of binding (*Figure 2Ci*). Unlike *Drosophila* tropomyosin Tm1A (*Hsiao et al., 2015*), which favors association near the actin filament pointed end, initial Cdc8 loading events had no preference for the barbed or pointed end of the actin filament, with binding patterns consistent with stochastic association (*Figure 3A–B*, *Figure 3—figure supplement 1*). Once an initial Cdc8 site was initiated, spreading of Cdc8 occurred toward both the barbed and pointed ends at similar rates (3.2 and 2.4 Cdc8 molecules sec$^{-1}$ µM$^{-1}$, respectively, p-value=0.037).

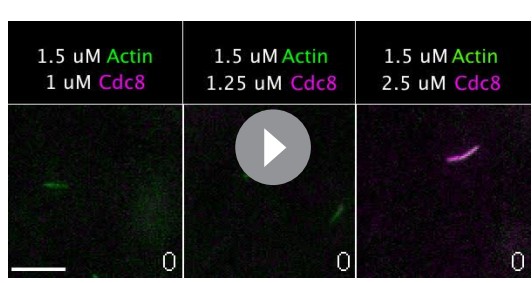

**Video 1.** Tropomyosin Cdc8 loads cooperatively onto F-actin, related to *Figure 1*. Two-color TIRF microscopy of 1.5 µM actin (Alexa-488 labeled) with a range of concentrations of tropomyosin Cdc8 (Cy5-labeled). Scale bar, 5 µm. Time in sec.

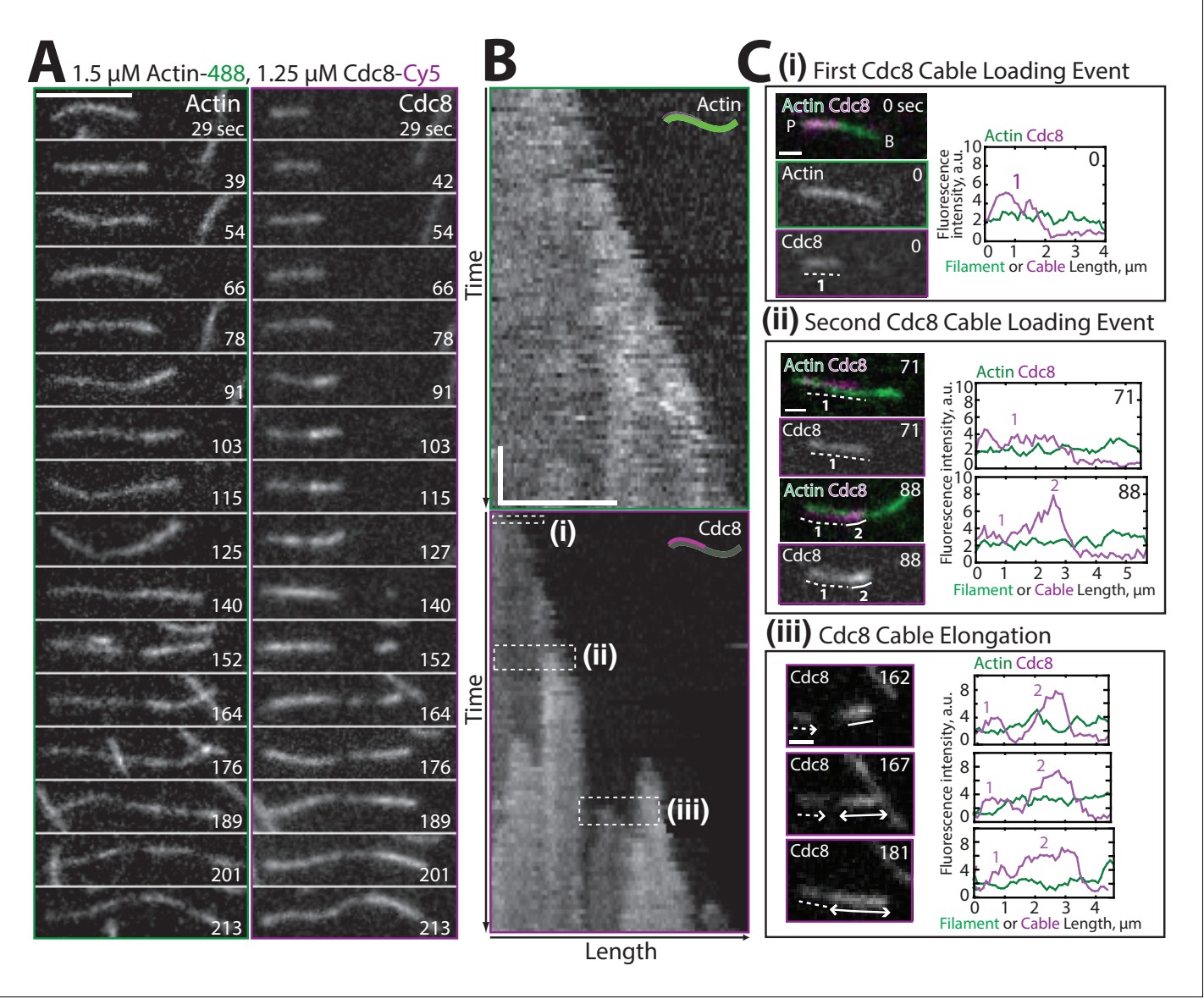

**Figure 2.** Two distinct Tropomyosin Cdc8 cables load cooperatively onto a single actin filament. (A–B) Two-color TIRFM of 1.5 μM Mg-ATP actin (15% Alexa 488) with 1.25 μM tropomyosin Cdc8 dimer (Cy5-labeled). (A) Timelapse of an elongating actin filament (left) and Cdc8 loading and spreading events (right). Scale bar, 5 μm. (B) Kymograph of the elongating actin filament and associated Cdc8 events. The first Cdc8 cable loading event (i), second Cdc8 cable loading event (ii), and Cdc8 cable spreading event (iii) are boxed. Scale bar, 5 μm. Time bar, 30 s. (C) Fluorescent images and corresponding fluorescence intensity line scans of actin (green) and Cdc8 (magenta) from the boxed regions in (B). Scale bar, 1 μm. (Ci) A single Cdc8 cable on an actin filament segment. The dotted line (1) marks one Cdc8 cable on the actin filament segment. (Cii) A second Cdc8 cable loading event on an actin filament. Dotted (1) and solid (2) lines mark the first and second tropomyosin cables on the actin filament segment. (Ciii) A spreading Cdc8 cable. Arrows denote spreading direction of first (dotted line, 1) and second (solid line (2) Cdc8 cables. Scale bars, 1 μm.

However, there was considerable variation in spreading rates in both directions, between ~1–6 Cdc8 molecules $sec^{-1}$ $μM^{-1}$ (*Figure 3C–D*). As the majority of Cdc8 molecules associated adjacent to a previously-bound Cdc8, these findings suggest that end-to-end binding is one key feature contributing to the high cooperativity of tropomyosin.

A cryo-electron microscopy structure of mammalian tropomyosin, as well as negative-stain electron microscopy of *S. pombe* Cdc8 has shown that a single actin filament accommodates two tropomyosin cables, one on each side of the helical actin filament (*Hanson and Lowy, 1963*;

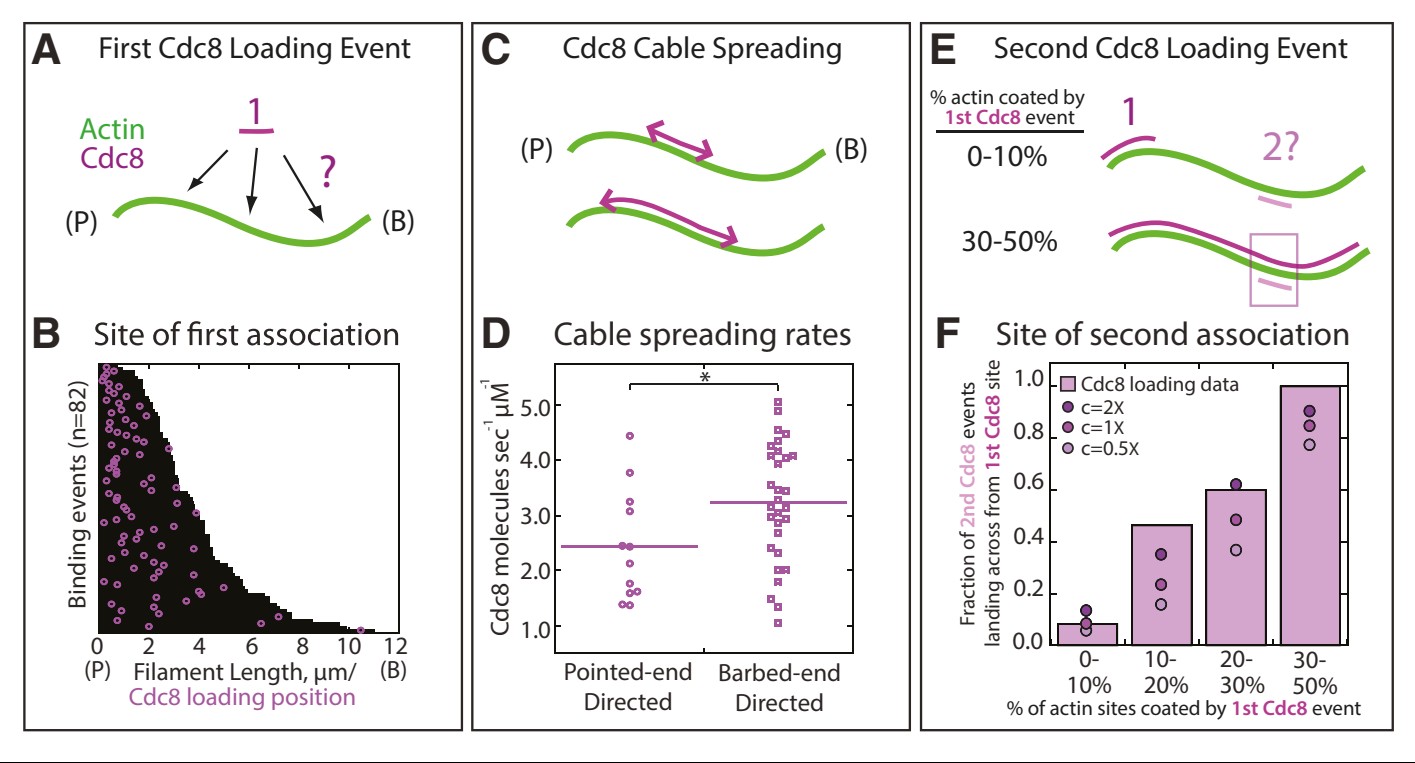

**Figure 3.** Quantification of loading and spreading of first and second Tropomyosin Cdc8 cables. Two-color TIRFM of 1.5 μM Mg-ATP actin (15% Alexa 488) with 1.25 μM tropomyosin Cdc8 dimer (Cy5-labeled). (**A**) Depiction of potential sites for the first Cdc8 loading event. The actin filament and Cdc8 molecule are depicted by green and purple lines, respectively. (**B**) Plot of the first Cdc8 association event (purple circles) on actin filaments (black lines), with F-actin pointed ends (P) aligned at the left. n = 82 events. (**C**) Depiction of Cdc8 cable spreading toward the barbed (B) and pointed (P) ends of actin filaments. (**D**) Spreading rates of Cdc8 cables toward the barbed or pointed end. Purple line denotes mean. Two-tailed t-test for data sets with equal variance yielded p-value *p=0.037. n > 12 elongation events. (**E**) Depiction of site of second Cdc8 loading event, which can occur at a naked actin site (top cartoon) or across from the first-bound Cdc8 cable (bottom cartoon). The larger percentage of the F-actin surface coated by the initial Cdc8 cable (1) increases the probability that the second Cdc8 cable (2) will associate across from a site already bound by Cdc8. (**F**) Plot of the fraction of second Cdc8 events that associate with a F-actin site already coated by Cdc8, binned by percentage of F-actin already coated by Cdc8 (light purple bars). n = 38 events. Modeling of predicted degree of association given no indirect cooperativity (c = 1X, purple circles), positive indirect cooperativity (c = 2X, dark purple circles), and negative indirect cooperativity (c = 0.5X, light purple circles).

The following figure supplement is available for figure 3:

**Figure supplement 1.** Tropomyosin Cdc8 first-binding events are consistent with random binding.

*Moore et al., 1970*; *Skoumpla et al., 2007*). In our TIRFM assays, we observed the loading of these two distinct Cdc8 cables on the same actin filament (*Figure 2Cii–iii*, *Video 2*). A second Cdc8 cable loading event was evident by a doubling in fluorescence intensity at sites of previously loaded Cdc8 cable (*Figure 2C*, right panels). The single pixel resolution in our TIRFM experiments is 100 nm, or a continuous stretch of ~5 Cdc8 molecules bound end-to-end. We assume that Cdc8 spreading events are the result of end-to-end binding between Cdc8 molecules. However, this resolution does not allow us to rule out the possibility of unconnected Cdc8 molecules binding near, but not immediately next to, previously loaded Cdc8 molecules. The doubling in Cdc8 fluorescence intensity upon association of the second Cdc8 cable allows us to explore the possibility of actin filament-mediated cooperative Cdc8 interactions.

We investigated whether the two distinct Cdc8 cables load independently, or whether there is a loading bias of the second Cdc8 cable to regions occupied by the first Cdc8 cable. In an initial Cdc8 loading event, a Cdc8 'seed' forms and elongates along one face of the actin filament. Second Cdc8 loading events can occur on the actin face opposite the first Cdc8 cable or on either face at a stretch of the actin filament free of Cdc8 (*Figure 3E*). As expected, if the first Cdc8 cable covered a higher

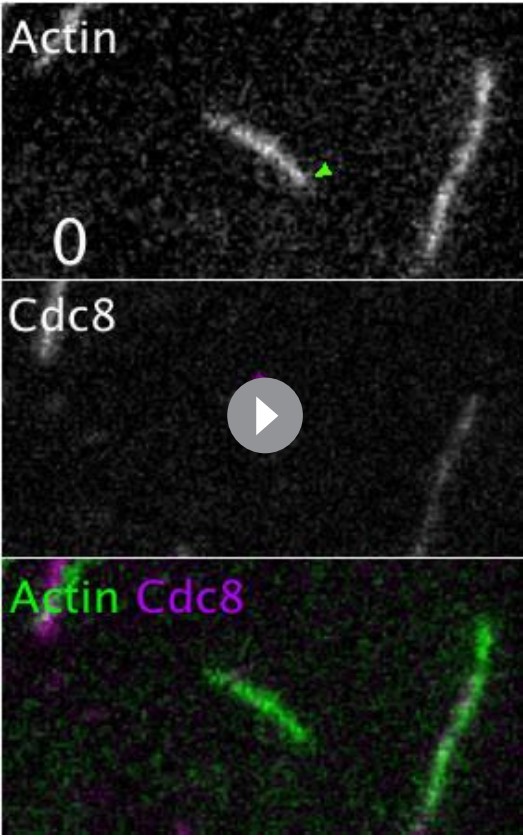

**Video 2.** Two distinct cables of Tropomyosin Cdc8 load onto a single actin filament, related to **Figure 2**. Two-color TIRF microscopy of 1.5 μM actin (Alexa-488 labeled) with 1.25 μM tropomyosin Cdc8 (Cy5-labeled). Green arrowhead indicates elongating F-actin barbed end. Purple arrowhead indicates the loading and spreading of Cdc8 cables. Scale bar, 5 μm. Time in sec.

fraction of the actin filament face, the second Cdc8 binding event was more likely to occur across from it (**Figure 3F**, purple bars). However, comparisons to a model of stochastic binding suggested that the observed binding opposite a first Cdc8 cable was higher than what was predicted (**Figure 3F**, c = 1X circles). The addition of an indirect cooperativity factor of 2 (doubling the likelihood of binding opposite a first Cdc8 cable) more closely replicated the data (**Figure 3F**, c = 2X circles). These findings suggest that although initial Cdc8 molecules bind stochastically on the actin filament, there is a bias for subsequent Cdc8 molecules to bind opposite previously-established Cdc8 stretches, indicating a potential role for indirect cooperativity (**Figure 1C**).

## Modeling of tropomyosin Cdc8 loading onto growing actin filaments

The preference of second Cdc8 loading events for already Cdc8-occupied regions described above is based on a statistical calculation from a limited number of second Cdc8 binding events (n = 37). To further address whether indirect cooperativity plays a role in tropomyosin Cdc8 loading, we created two variations of a lattice simulation describing Cdc8 loading onto a growing actin filament (**Figure 4**). The first model described Cdc8 loading with only end-to-end cooperativity (w) (**Figure 4Bi**), while the second described Cdc8 loading with both end-to-end (w) and indirect cooperativity (c) across the actin filament (**Figure 4Ci**). We sought to determine whether either model was sufficient to reproduce the complex behavior observed experimentally (**Figure 4Ai–iii**). To select values of w, c and $k_{off}$, we first fixed c, and varied $k_{off}$ and w until generating (1) the best match to the experimental data in **Figure 1B** as $k_{on}$ is varied, (2) a similar $k_{off}$ to single molecule experimental measurements (**Figure 4—figure supplement 1**), and (3) a similar initial Cdc8 binding time to experimental measurements (**Figure 3—figure supplement 1**). In the first model, cooperativity occurred only via end-to-end binding (w) (**Figure 4Bi**). A kymograph generated using those parameters replicated many of the observed Cdc8 loading characteristics, specifically the frequent starts and stops and variation in spreading rate (**Figure 4Bii**). However, a much lower fraction of F-actin was doubly coated with Cdc8 in the model (**Figure 4Biii**) compared to the experimental data (**Figure 4Aiii**). Therefore, end-to-end cooperative binding alone does not fully account for the experimentally observed Cdc8 loading behavior. We therefore added a factor of positive indirect cooperativity (c) to the model (**Figure 4Ci**). An indirect cooperativity (c) value of 1.25 most closely replicated our experimental data (**Figure 4C**). This model replicated the observed dynamics of Cdc8 loading (**Figure 4Cii**), and also replicated the observed bias towards double-coating of actin filaments by Cdc8 (**Figure 4Ciii**, compare to **Figure 4Aiii**). These findings, along with the second Cdc8 binding event data (**Figure 3F**), are consistent with a role for indirect cooperativity in the high overall cooperativity of Cdc8.

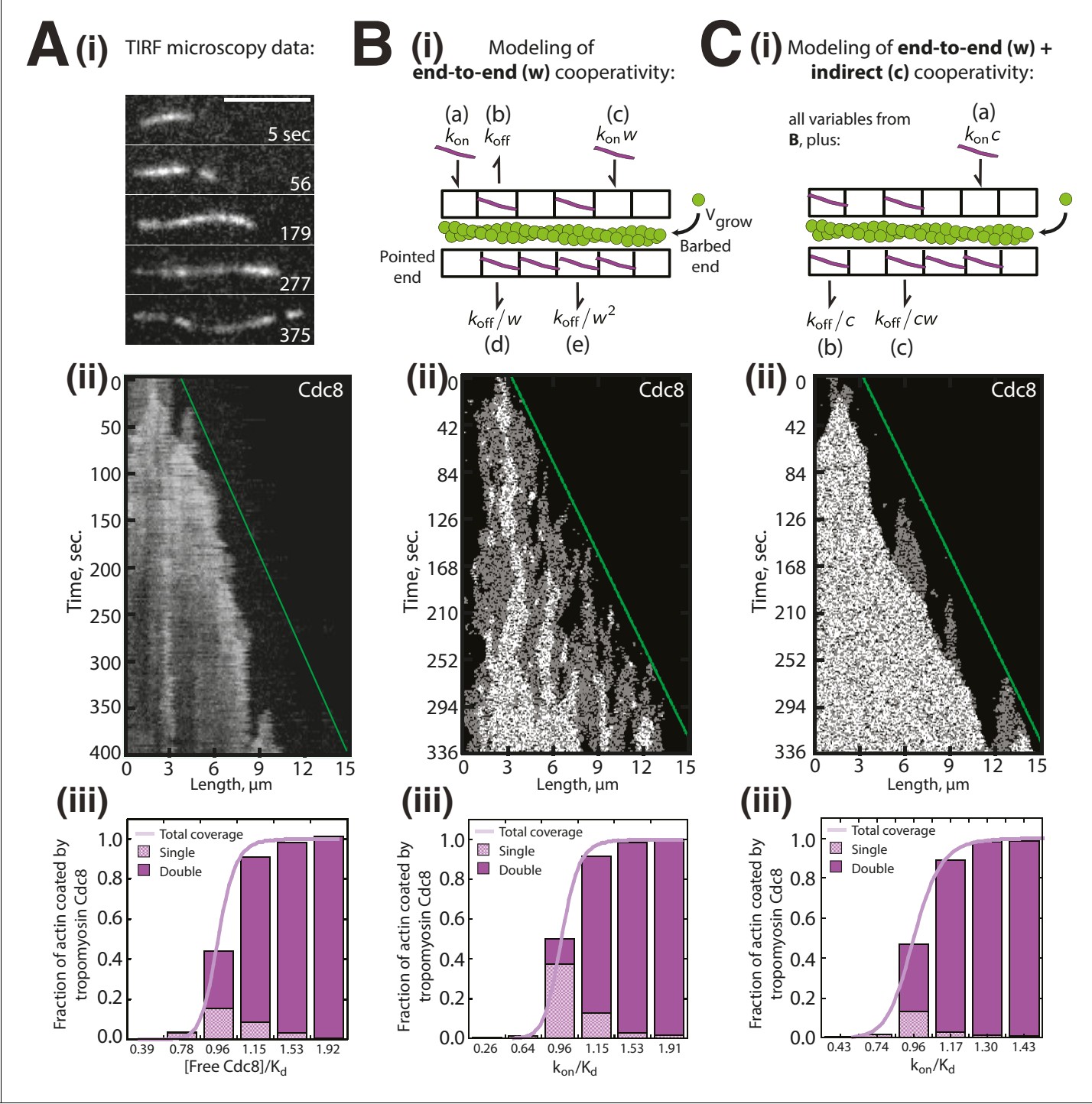

**Figure 4.** Modeling of Tropomyosin Cdc8 dynamics on growing actin filaments. (**Ai-iii**) Two-color TIRFM of 1.5 µM Mg-ATP actin (15% Alexa 488) with 1.25 µM tropomyosin Cdc8 dimer (Cy5-labeled). (**Ai-ii**) Timelapse and corresponding kymograph of Cdc8 loading and spreading. The green line indicates the actin filament barbed end. Scale bar, 5 µm. (**Aiii**) Quantification of the fraction of F-actin coated by one (Single, checkered purple) or two (Double, solid purple) Cdc8 cables. Total coverage (purple line) is from initial quantification in *Figure 1B*. Hill=14.6. n = 2 reactions. (**Bi-iii**) Modeling of Cdc8 association with an actin filament with exclusively end-to-end interactions. (**Bi**) Lattice model schematic with parameters for actin elongation ($v_{grow}$), rates of association ($k_{on}$, (**a**)) or dissociation ($k_{off}$, (**b**)) of single Cdc8 molecules with the actin filament, and rates of association ($k_{on}*w$ (**c**), $k_{on}*w^2$) and dissociation ($k_{off}/w$ (**d**), $k_{off}/w^2$ (**e**)) at sites within a Cdc8 cable. (**Bii**) Kymograph of simulated loading and spreading of modeled Cdc8 under parameters in (**Bi**). The green line indicates the actin filament barbed end. (**Biii**) Quantification of simulated data from end-to-end cooperativity model. Hill=14.9. (**Ci-iii**) Modeling of Cdc8 association with an actin filament that includes both end-to-end interactions and indirect cooperativity. (**Ci**)

*Figure 4 continued on next page*

Figure 4 continued

Schematic of lattice model, which includes all parameters from (B) as well as additional parameters added for (C): rates of association ($k_{on}$*c, (a)) and dissociation ($k_{off}$/c, (b)) of Cdc8 molecules across from a site already bound by a Cdc8 molecule, and rates of association ($k_{off}$*cw) and dissociation ($k_{off}$/cw (c)) of Cdc8 molecules within a cable and across from an already-bound Cdc8. (Cii) Kymograph of simulated loading and spreading of modeled Cdc8 under parameters in (Ci). (Ciii) Quantification of simulated data from end-to-end with indirect cooperativity model. Hill=13.4.

The following figure supplement is available for figure 4:

**Figure supplement 1.** Residence time of tropomyosin Cdc8 on actin filaments.

## Tropomyosin Cdc8 is actively displaced from F-actin bundled by fimbrin Fim1

Tropomyosin Cdc8's high cooperativity allows it to rapidly coat actin filaments. This property could be physiologically important as it allows tropomyosins to quickly define the functional composition of an F-actin network by associating with F-actin and influencing the subsequent association of other ABPs (*Gunning et al., 2015*; *Johnson et al., 2014*; *Tojkander et al., 2011*). In fission yeast, Cdc8 inhibits several actin patch ABPs from binding F-actin, including ADF/cofilin Adf1 and myosin-I Myo1 (*Clayton et al., 2010*; *Skau and Kovar, 2010*). Though Cdc8's presence at the contractile ring and actin cables likely prevents the unwanted association of several types of ABPs at those F-actin networks, opposing mechanisms must be in place to prevent Cdc8 from associating with actin patches. Cdc8's association with actin patches is regulated by competition with the F-actin crosslinking protein fimbrin Fim1 (*Skau and Kovar, 2010*), which localizes predominantly to actin patches in fission yeast (*Nakano et al., 2001*; *Wu et al., 2001*). Though Cdc8 does not associate with actin patches normally, Cdc8 localizes to actin patches in the absence of fimbrin (*fim1-1Δ* cells) (*Skau and Kovar, 2010*). In addition, Fim1 prevents Cdc8 from binding F-actin in steady state in vitro bulk sedimentation assays (*Skau and Kovar, 2010*). However, the mechanism by which Fim1 prevents the association of Cdc8 with F-actin is unclear. We suspected that Fim1 could inhibit the initial ability of Cdc8 to load onto F-actin. Alternatively, Fim1 could actively facilitate the dissociation of Cdc8 from actin filaments.

To differentiate between these and other potential mechanisms, we utilized three-color TIRFM to follow the assembly of labeled actin in the presence of labeled Fim1 and Cdc8 to simultaneously observe their associations with F-actin (*Figure 5*). Cdc8 initially coated individual actin filaments (*Figure 5B*, first panel), with loading and spreading patterns similar to those observed in the absence of Fim1. However, as actin filaments became bundled by Fim1 (*Figure 5A,C*), the amount of Fim1 associated with the actin filaments increased and Cdc8 was actively removed from the bundled region (*Figure 5B–D*, *Video 3*). We observed Cdc8 displacement to some extent from every Fim1-mediated bundle formed. Kymographs of elongating actin filaments bundled by Fim1 showed that Cdc8 displacement occurred in a cooperative manner at the bundled site, where Cdc8 cables appeared to be 'stripped' away from initial sites of dissociation (*Figure 5E*). Fim1 fluorescence intensity increased at F-actin bundles compared to single actin filaments (*Figure 5F*), while Cdc8 intensity decreased at F-actin bundles (*Figure 5G*), indicating that Cdc8 displacement by Fim1 occurred preferentially at sites of F-actin bundling (*Figure 5G*). These findings demonstrate that Fim1-mediated bundling displaces Cdc8 from the actin network. Additionally, as Cdc8 appears to be rapidly 'stripped' from a few initial dissociation points, it is possible that the cooperativity of Cdc8 assists not only its assembly onto filaments but also its rapid displacement specifically from F-actin networks bundled by Fim1.

## Tropomyosin Cdc8 inhibits initial ADF/cofilin Adf1 binding to actin filaments

ADF/cofilin Adf1 is an F-actin severing protein that localizes predominantly to actin patches and is required for proper actin patch dynamics (*Nakano et al., 2001*; *Nakano and Mabuchi, 2006*), but also localizes to the contractile ring and is critical for its assembly (*Chen and Pollard, 2011*). Adf1 is an important F-actin network disassembly factor that severs actin filaments and allows recycling of the actin monomers (*Andrianantoandro and Pollard, 2006*; *Chen and Pollard, 2013*). Competition

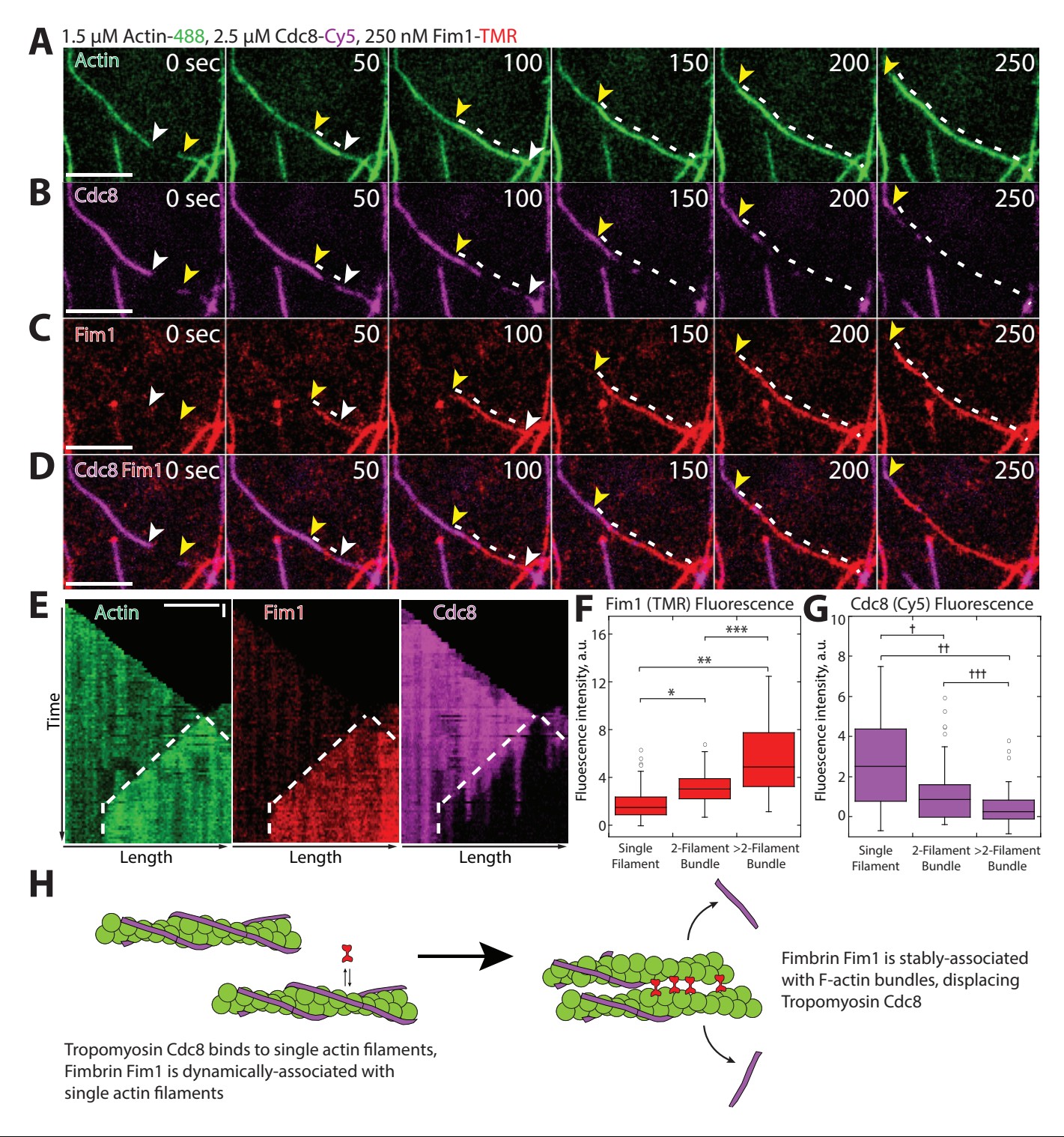

**Figure 5.** Fimbrin Fim1-mediated bundling induces cooperative removal of Tropomyosin Cdc8. (A–E) Timelapse of three-color TIRFM of 1.5 μM Mg-ATP actin (15% Alexa 488-labeled) with 2.5 μM tropomyosin Cdc8 (Cy5-labeled) and 250 nM fimbrin Fim1 (TMR-labeled). Arrowheads and dotted line mark actin filament barbed ends and bundled region, respectively. (E) Kymographs of actin, Fim1, and Cdc8 during bundle formation. Dotted lines indicate the bundled region. Scale bars, 5 μm. Time bar, 30 s. (F–G) Box plots of the amount of Fim1-TMR (F) or Cdc8-Cy5 (G) fluorescence on single actin filaments, two-filament bundles, or bundles containing more than two filaments. Open circles indicate outliers. Two-tailed t-tests for data sets with unequal variance yielded p-values: *p-value=$4.67 \times 10^{-10}$, **p-value=$4.74 \times 10^{-11}$, ***p-value=$1.38 \times 10^{-5}$, †p-value=$4.52 \times 10^{-7}$, ††p-value=$1.16 \times$

*Figure 5 continued on next page*

*Figure 5 continued*

$10^{-12}$, [†††]p-value=0.01. n > 42 measurements. (**H**) Cartoon model of how Fim1 and Cdc8 affect each other's association with single and bundled actin filaments.

between tropomyosins and ADF/cofilins in many systems has been well-established (*Bernstein and Bamburg, 1982*; *DesMarais et al., 2002*; *Kuhn and Bamburg, 2008*; *Ono and Ono, 2002*). In fission yeast, Adf1-mediated severing is decreased in the presence of Cdc8 (*Skau and Kovar, 2010*). We used multi-color TIRFM to investigate whether Cdc8 inhibits Adf1-mediated severing by decreasing the initial association of Adf1 with F-actin, or by another mechanism. Wild-type Adf1 labeled with a cysteine-reactive dye was not observed to associate with F-actin (data not shown). We therefore engineered an Adf1 labeling mutant (C12S, C62A, D34C) for use in TIRFM, based on a similar strategy used for budding yeast ADF/cofilin Cof1 (*Figure 6—figure supplement 1*, Methods) (*Suarez et al., 2011*). The TMR-labeled Adf1 mutant is less active than wild-type Adf1 (*Figure 6—figure supplement 1A*), but was observed to bind F-actin in a cooperative manner and sever actin filaments at boundaries of Adf1 bound/unbound regions (*Figure 6—figure supplement 1B and C*), characteristics previously observed for ADF/cofilins from a variety of organisms including fission yeast (*Andrianantoandro and Pollard, 2006*; *Hayakawa et al., 2014*; *De La Cruz, 2009*; *Michelot et al., 2007*).

At high Adf1 concentrations (5 μM), the majority of F-actin was rapidly coated with Adf1 (*Figure 6A*, *Figure 6—figure supplement 1B*). However, in the presence of Cdc8, little Adf1 was observed to initially associate with actin filaments (*Figure 6A–C*, *Video 4*). Over time, Adf1 began to load near the pointed end of Cdc8-associated actin filaments in small 'clusters' that then spread cooperatively along the filament (*Figure 6B*). Importantly, though Cdc8 affected the initial association ($k_{on}$) of Adf1 with actin filaments, the dissociation rate ($k_{off}$) of Adf1 from F-actin was unaffected by Cdc8 (*Figure 6D*). These findings are supported by three-color TIRFM imaging of the assembly of labeled actin in the presence of labeled Adf1 and Cdc8. Labeled Cdc8 and Adf1 show mutually exclusive localization on actin filaments (*Figure 6E*). Initially, the majority of actin filaments are coated with Cdc8. Over time, Adf1 puncta arise and displace Cdc8 from those regions. Adf1 domains then spread in a cooperative manner, further displacing Cdc8 (*Figure 6F*). Cdc8 continues to associate with the elongating actin filament barbed end, where Adf1 is not yet associated, but is ultimately displaced by Adf1 over time. As Adf1 preferentially associates with ADP-F-actin (*Andrianantoandro and Pollard, 2006*), we suspect that the transition from ADP-Pi- to ADP-actin increases the $k_{on}$ of Adf1 for ADP-bound F-actin, allowing a few Adf1 molecules to associate with the actin filament. Adf1 then cooperatively spreads from these regions of association, and displaces Cdc8. Collectively, these data suggest that while Cdc8 inhibits the initial binding of Adf1 to actin filaments, Adf1 is capable of associating and ultimately displacing segments of Cdc8 (*Figure 6G*).

## Fimbrin Fim1 and ADF/cofilin Adf1 synergize to quickly generate a dense F-actin network

Within an actin patch, a dense array of Arp2/3 complex-mediated branched F-actin provides the network architecture that propels newly-generated endocytic vesicles inward (*Collins et al., 2011*; *Young et al., 2004*). We previously postulated that fimbrin Fim1 may be required for proper actin patch motility indirectly, by inhibiting tropomyosin Cdc8 association and therefore allowing ADF/cofilin Adf1-mediated severing to occur (*Skau and Kovar, 2010*). However, a separation of function Fim1 mutant that can bind but not bundle F-actin, and retains the ability to displace Cdc8 from F-actin, still shows abnormal actin patch dynamics (*Skau et al., 2011*). Therefore Fim1 has additional roles aside from

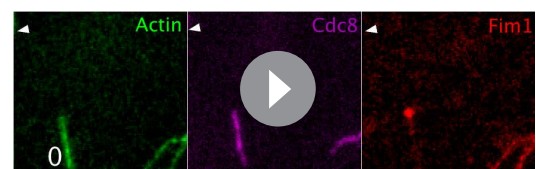

**Video 3.** Fimbrin Fim1 actively displaces Tropomyosin Cdc8 from F-actin bundles, related to *Figure 5*. Three-color TIRF microscopy of 1.5 μM actin (Alexa-488 labeled) with 250 nM fimbrin Fim1 (TMR-labeled) and 2.5 μM tropomyosin Cdc8 (Cy5-labeled). Arrowheads indicating two distinct F-actin elongating barbed ends. Scale bar, 5 μm. Time in sec.

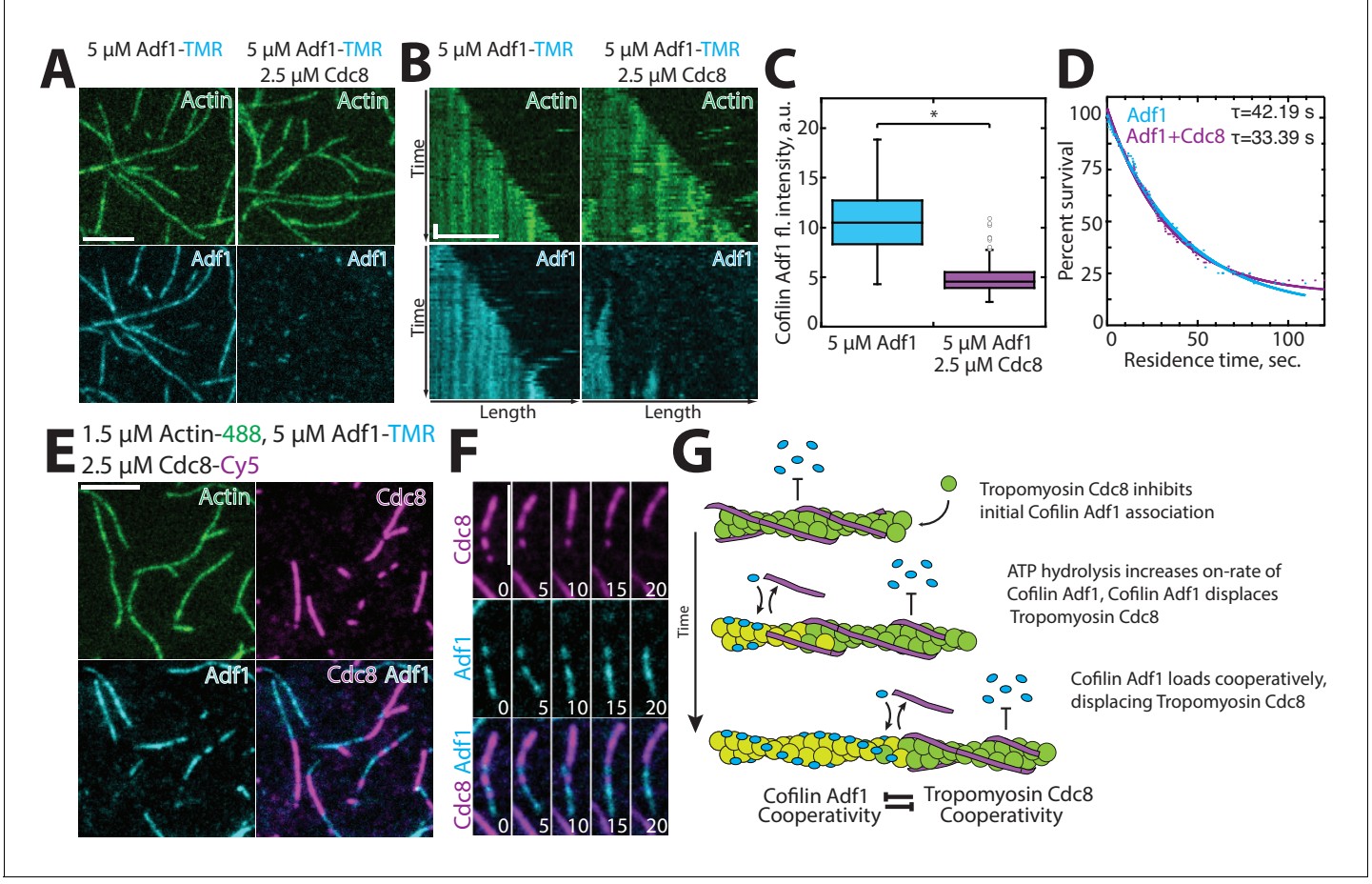

**Figure 6.** Tropomyosin Cdc8 inhibits initial binding of ADF/cofilin Adf1 to actin filaments. (**A–C**) Two-color TIRFM of 1.5 µM Mg-ATP actin (15% Alexa 488 labeled) with 5 µM ADF/cofilin Adf1 (TMR-labeled) in the presence or absence of 2.5 µM tropomyosin Cdc8 (unlabeled). (**A**) Micrograph of Adf1 association with actin filaments in the absence or presence of Cdc8. (**B**) Kymograph of an elongating actin filament (top) and associated Adf1 (bottom) in the absence or presence of unlabeled Cdc8. Scale bar, 5 µm. Time bar, 30 s. (**C**) Box plot of average Adf1 fluorescence intensity on actin filaments in the absence or presence of unlabeled Cdc8. Open circles indicate outliers. Two-tailed t-test for data sets with unequal variance yielded *p-value: $6.77 \times 10^{-82}$. n ≥ 145 measurements. (**D**) Residence time of single Adf1 (TMR-labeled) molecules in the absence or presence of unlabeled Cdc8. n ≥ 81 events. (**E–F**) Three-color TIRFM of 1.5 µM Mg-ATP actin (15% Alexa 488 labeled) with 5 µM Adf1 (TMR-labeled) and 2.5 µM Cdc8 (Cy5-labeled). (**E**) Micrograph of actin filaments associated with Adf1 and Cdc8. Scale bar, 5 µm. (**F**) Timelapse of Adf1 association with an actin filament, and subsequent dissociation of Cdc8. Scale bar, 5 µm. Time in sec. (**G**) Cartoon model of how Cdc8 and Adf1 affect each other's association with F-actin.

The following figure supplement is available for figure 6:

**Figure supplement 1.** Characterization of Cofilin Adf1 labeling mutant.

protecting actin patches from Cdc8. Therefore, we examined the relationship between Fim1 and Adf1 utilizing multi-color TIRFM with labeled proteins (*Figure 7*). Unlike the mutually-exclusive localization of Cdc8 and Adf1 (*Figure 6E*), Fim1 and Adf1 co-localized at both single actin filaments and multi-filament bundles throughout the TIRF chamber (*Figure 7B*). However, distinct Fim1 or Adf1 domains could still be observed, and severing often occurred at these interfaces (*Figure 7C*, yellow line), likely because of sharp changes in Adf1 density and/or actin filament flexibility between those two regions (*Elam et al., 2013*; *McCullough et al., 2008*; *Suarez et al., 2011*). Consistent with this finding, the Adf1 severing rate was increased in the presence of Fim1 (*Figure 7F*). As the severing rate could only be directly observed on single actin filaments and two-filament bundles, our measured severing rate in the presence of Fim1 is likely underreported (*Figure 7F*, indicated by #). Severed filaments were quickly incorporated into nearby bundles, resulting in an increase in Fim1-mediated bundling in the

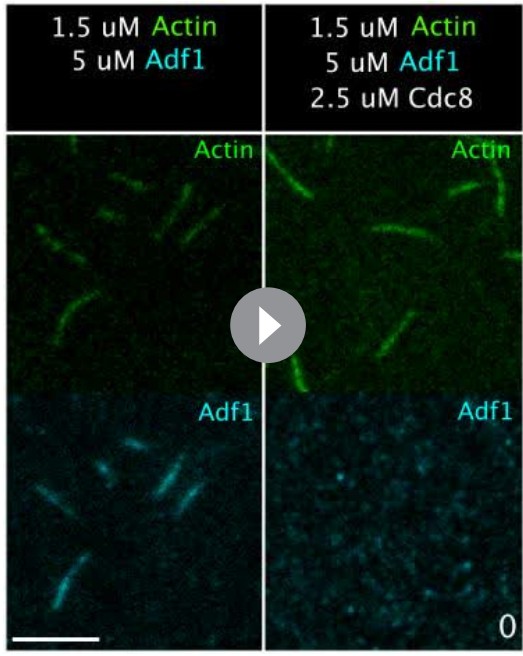

**Video 4.** Tropomyosin Cdc8 prevents initial association of Cofilin Adf1 with F-actin. Two-color TIRF microscopy of 1.5 μM actin (Alexa-488 labeled) with 5 μM cofilin Adf1 (TMR-labeled). Scale bar, 5 μm. Time in sec.

presence of Adf1 (*Figure 7D*, *Video 5*). Fim1-mediated bundles became extremely large and dense in the presence of Adf1 (compare *Figure 7B* to 7A), as each severing event resulted in the formation of a new elongating barbed end. This rapid generation of barbed ends yielded a ~11 fold change in actin fluorescence after 300 s (*Figure 7E*, solid lines), compared to a ~2 fold change in experiments lacking Adf1 (*Figure 7E*, dashed lines). Together, these findings support a potential role for Adf1 as both a disassembly factor (via severing) and an assembly factor (via generation of new barbed ends). We speculate that the presence of Fim1 on actin patches may be important not only for exclusion of Cdc8 from the network but also for enhancement of Adf1-mediated severing via generation of single filament/bundle interfaces (*Figure 7G*).

## Fimbrin Fim1 and ADF/cofilin Adf1 work together to displace Tropomyosin Cdc8 from actin filaments

Our findings and previous work have suggested that the synergistic activities of fimbrin Fim1 and ADF/cofilin Adf1 may serve to inhibit the association of tropomyosin Cdc8 with F-actin (*Skau and Kovar, 2010*). We tested this possibility using four-color TIRFM to examine how Fim1, Adf1, and Cdc8 collectively affect each other's interactions with F-actin (*Figure 8*). As in the previous experiments with only Fim1 and Adf1 (*Figure 7*), a dense, bundled F-actin network was formed, containing filaments coated with both Adf1 and Fim1 (*Figure 8A*). Cdc8, on the other hand, was rapidly dissociated from nearly every actin filament in the chamber (*Figure 8A–C*, *Video 6*). Closer observation revealed individual growing actin filaments initially coated with Cdc8 (*Figure 8D*, right panel). These filaments are often severed at single/bundled actin filament boundaries (*Figure 8Di*, yellow line), and the severing events created new actin filament barbed ends that were encompassed into F-actin bundles, where Cdc8 was displaced by Fim1 (*Figure 8Dii*). We did not observe Adf1 directly displacing Cdc8, and therefore we suspect that Adf1's primary role in displacing Cdc8 is by rapidly generating new actin filament barbed ends that are encompassed into F-actin bundles mediated by Fim1 (*Figure 8E*).

## Discussion

### Tropomyosin Cdc8 cooperativity

Our work demonstrates that cooperativity is an important characteristic for multiple aspects of F-actin network formation, organization, and ABP sorting. We show that fission yeast tropomyosin Cdc8 binds extremely cooperatively to F-actin (*Figure 1*). Though tropomyosins from other organisms have been labeled for use in TIRFM (*Hsiao et al., 2015*; *Schmidt et al., 2015*), ours is the first study to observe two distinct tropomyosin cables associating with a single actin filament (*Figure 2*). This high degree of resolution has provided us the unique opportunity to differentiate between the contributions of (1) end-to-end association and (2) indirect interactions via the actin filament in tropomyosin's cooperative binding to F-actin (*Figure 4*).

End-to-end binding has been considered to be the primary source of tropomyosin's cooperativity. Mutations in the N- or C-terminal domains of Cdc8 affect its ability to bind cooperatively and polymerize on an actin filament (*East et al., 2011*). However, it has been suggested that other

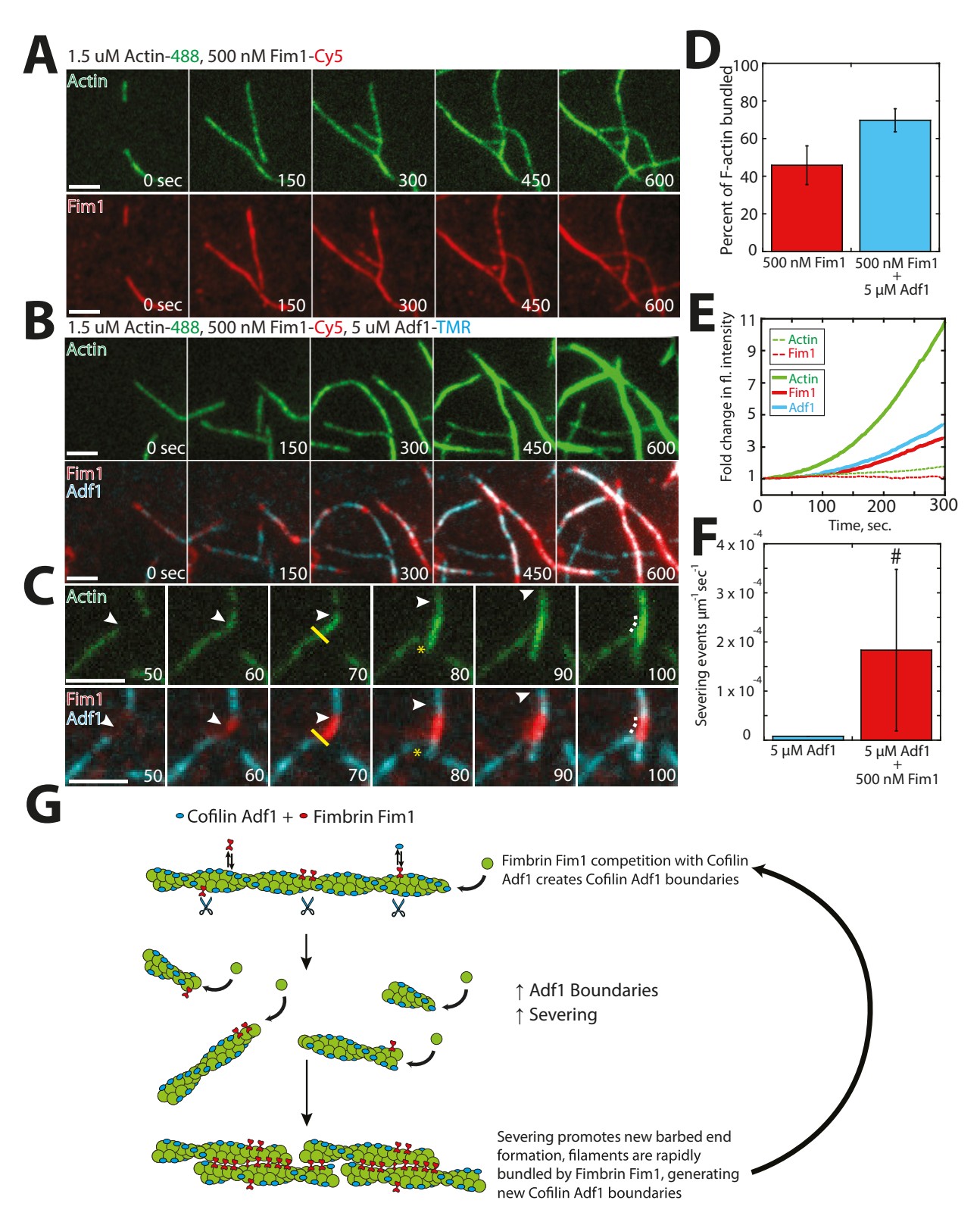

**Figure 7.** Fimbrin Fim1 and ADF/cofilin Adf1 competition generates a dense F-actin network. (**A**) Timelapse of two-color TIRFM of 1.5 μM Mg-ATP actin (15% Alexa 488-labeled) with 500 nM fimbrin Fim1 (Cy5-labeled). (**B**) Timelapse of three-color TIRFM of 1.5 μM Mg-ATP actin (15% Alexa 488-labeled) with 500 nM Fim1 (Cy5-labeled) and 5 μM ADF/cofilin Adf1. (**C**) Timelapse showing severing at the boundary between a Fim1-mediated bundle and a single actin filament. White arrow indicates the elongating actin filament barbed end. A yellow line and asterisk indicate the severing site and

*Figure 7 continued on next page*

*Figure 7 continued*

severing event, respectively. (**D**) Quantification of percent of total actin filaments bundled with Fim1 alone or Fim1 and Adf1. Error bars represent standard deviation of the mean; n = 2 reactions. (**E**) Fold-change over time in total fluorescence intensity for either actin (dotted green line) and Fim1 (dotted red line) in the absence of Adf1, or for actin (solid green line), Fim1 (solid red line), and Adf1 (solid blue line) in the presence of Adf1. (**F**) Severing rate of high concentrations of Adf1 alone or in the presence of Fim1. # indicates under-reporting, as severing events could not be measured on dense bundles. Error bars represent standard deviation of the mean; n = 2 reactions. (**G**) Cartoon model of how Adf1 and Fim1 influence each other's interactions with actin and affect F-actin network formation.

interactions between tropomyosin and the actin filament may also be involved in tropomyosin cooperativity as end-to-end attachments between tropomyosin molecules tend to be rather weak (*Sousa and Farah, 2002*) and muscle tropomyosin with impaired ability to associate end-to-end is still cooperative on actin filaments (*Willadsen et al., 1992*; *Tobacman, 2008*). Our findings suggest that, though end-to-end binding is a key factor in tropomyosin cooperativity, indirect interactions via the actin filament may also be important for enhancing tropomyosin coating of F-actin. Our results suggest that a small increase in indirect cooperativity (increase in binding affinity of Cdc8 by a factor of ~2) is sufficient to account for Cdc8's high cooperativity and observed loading characteristics (*Figure 4C*). The precise interactions within the actin filament that allow for this cooperativity remain to be determined. However, due to the logarithmic relationship between free energy differences and equilibrium constants, a factor of 2 increase in binding affinity corresponds to a very small increase in free-energy stabilization of $-k_BT\ln(2)=$~0.4 kcal/mol. Hence, if this indirect cooperativity is due to a structural change in the actin filament, we expect that change to be quite subtle.

Tropomyosin is known to increase the persistence length of F-actin (*Fujime and Ishiwata, 1971*; *Isambert et al., 1995*). As a result, the stabilizing effect of an initial Cdc8 binding to one side of the actin filament may favor binding of a second Cdc8 to the opposing side. Additionally, if the rigidity provided by initial Cdc8 binding is propagated slightly further up or down the actin filament, more tropomyosin molecules could associate with the actin filament as a result of an increase in F-actin rigidity. More complex mammalian cells express >40 tropomyosin isoforms that vary in affinity for F-actin and degree of cooperativity (*Pittenger et al., 1994*; *Schevzov et al., 2011*), suggesting that these characteristics may differentially affect their distinct cellular roles and ability to sort to different F-actin networks (*Gunning et al., 2015*).

## Tropomyosin cooperativity and ABP sorting

How does this high degree of cooperativity affect tropomyosin's influence on the activity of other ABPs? The nature of tropomyosin as a strand-like, end-to-end associated protein makes it an ideal F-actin 'gatekeeper' (*Gunning et al., 2008*, *2015*). Individual tropomyosin Cdc8 molecules associate poorly with actin filaments. However, once a Cdc8 'seed' has been initiated, end-to-end binding and potential indirect interactions promote Cdc8's rapid coating of F-actin. The avidity generated from multiple F-actin-Cdc8 and Cdc8-Cdc8 interactions regulates the binding of other ABPs such as ADF/cofilin (*Figure 6A–B*). In addition, this large number of interactions between a Cdc8 cable and the actin filament could allow a Cdc8 cable to remain associated despite perturbations along the actin filament that may result from the association of other ABPs with the actin filament. However, these same cooperative characteristics also make Cdc8 easily removable from actin filaments once a threshold of perturbations has been bypassed. For example, at regions of high fimbrin Fim1 association (F-actin bundles), Cdc8's end-to-end associations allow it to rapidly peel away from those actin filaments (*Figure 5B*). In addition, the poor ability of

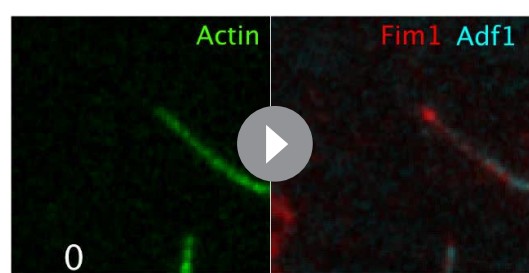

**Video 5.** Fimbrin Fim1 and Cofilin Adf1 synergize to generate a dense F-actin network, related to *Figure 7*. Three-color TIRF microscopy of 1.5 µM actin (Alexa-488 labeled) with 500 nM fimbrin Fim1 (Cy5-labeled) and 5 µM cofilin Adf1 (TMR-labeled). Arrowheads indicate sites of severing. Scale bar, 5 µm. Time in sec.

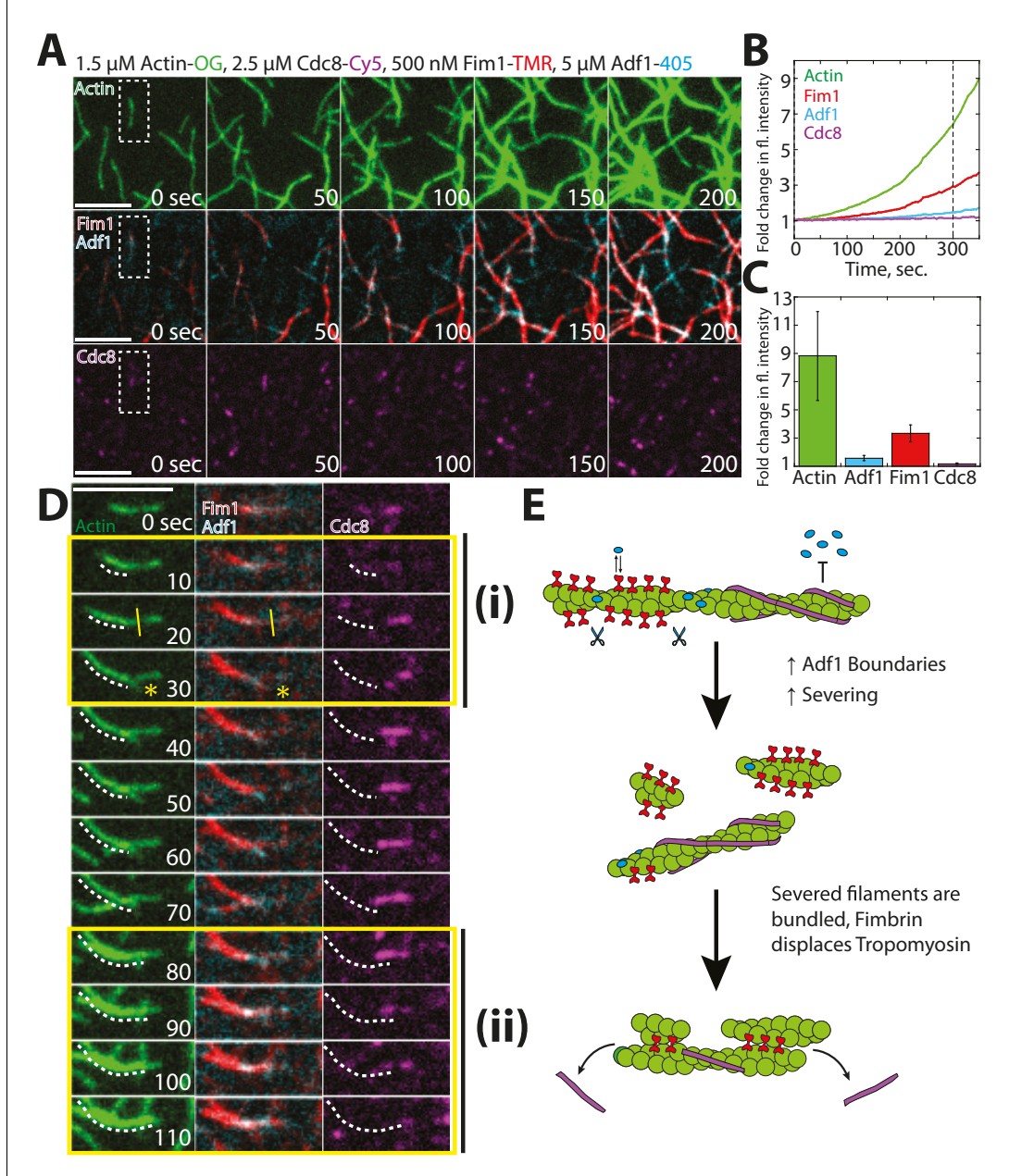

**Figure 8.** Competitive interactions between Cofilin Adf1 and Fimbrin Fim1 result in rapid displacement of Tropomyosin Cdc8 from F-actin networks. (A–D) Four-color TIRFM of 1.5 μM Mg-ATP actin (15% Alexa 488 labeled) with 2.5 μM tropomyosin Cdc8 (Cy5-labeled), 500 nM fimbrin Fim1 (TMR-labeled), and 5 μM cofilin Adf1 (Alexa 405-labeled). (A) Timelapse of F-actin network generation in the presence of actin (green), Fim1 (red), Adf1 (cyan), and Cdc8 (magenta). (B) Fold-change over time in fluorescence intensity of actin (green line), Fim1 (red line), Adf1 (blue line), and Cdc8 (purple line). (C) Fold change in fluorescence intensity of each ABP after 300 s. Error bars represent standard deviation of the mean. n = 2 independent experiments. (D) Enlargement of the area within the dotted box in (A). Fim1 and Adf1 synergize to create a dense F-actin bundle (white dotted line), while Cdc8 associates with a single actin filament. Severing occurs at the boundary of the single actin filament and F-actin bundle (i, yellow asterisk), creating a new elongating F-actin barbed end. The bundle extends to incorporate the single actin filament and Cdc8 is displaced (ii). Scale bar, 5 μm. (E) Cartoon model of events occurring in (D).

**Video 6.** Fimbrin Fim1 and Cofilin Adf1 work together to displace Tropomyosin Cdc8 from the F-actin network. Four-color TIRF microscopy of 1.5 µM actin (Alexa 488-labeled) with 500 nM fimbrin Fim1 (TMR-labeled), 2.5 µM tropomyosin Cdc8 (Cy5-labeled) and 5 µM cofilin Adf1 (Alexa 405-labeled). Scale bar, 5 µm. Time in sec.

individual Cdc8 molecules to bind to an actin filament makes it unlikely to be able to displace other ABPs once they're bound to an actin filament, resulting in Cdc8's complete exclusion from certain F-actin networks, such as actin patches. We suspect that Fim1's long residence time on F-actin bundles enhance its ability to create regions of high Fim1 occupancy that displace Cdc8, suggesting that ABP dynamics (on-/off-rate) may partially dictate the ability of an ABP to compete. An ABP's dynamics could be driven by its intrinsic biochemical properties (*Winkelman et al., 2016*), post-translational modifications (*Miao et al., 2016*), or mechanical stresses within the F-actin network (*Srivastava and Robinson, 2015*), resulting in the fine-tuning of ABP sorting by many factors.

Another impact of Cdc8's high cooperativity may be that only a slight bias toward certain actin filaments could generate an 'all-or-nothing' sorting toward those networks. Several studies have suggested that the actin assembly factor may be the key to generating this initial bias (*Kovar et al., 2011*). In fission yeast, altering formin localization in the cell results in the relocalization of both acetylated and unacetylated forms of Cdc8 (*Johnson et al., 2014*). This bias may be the result of formin-induced conformational changes that could be propagated down the actin filament (*Bugyi et al., 2006*; *Papp et al., 2006*). Future work will involve deciphering potential 'initiating' signals for ABP sorting and their effects on ABP cooperativity and competition.

## Competition and boundary generation

Our work also addresses the importance of ABP competition in network organization and ABP sorting. An outstanding question is how ADF/cofilin, a protein that severs at low concentrations, can be present at high concentrations in the cell and yet still rapidly disassemble F-actin networks. Other ABPs, such as Aip1 (*Gressin et al., 2015*; *Nadkarni and Brieher, 2014*), Coronin (*Jansen et al., 2015*), and Twinfilin (*Johnston et al., 2015*) have been found to enhance ADF/cofilin-mediated severing via multiple mechanisms, including potential side-binding by Aip1 (*Chen et al., 2015*). Our study additionally implicates side-binding ABPs not involved in severing (fimbrin Fim1) as important for the enhancement of ADF/cofilin Adf1 severing, likely by generating ADF/cofilin boundaries or changes in flexibility that result from Fim1-mediated bundle formation (*Figure 7C*). The idea of competition between ADF/cofilin and other factors as important for enhancing ADF/cofilin-mediated severing is an exciting area of investigation (*Elam et al., 2013*), and future work remains to determine how different ABPs regulate ADF/cofilin severing to different extents on different F-actin networks.

In addition, our study suggests an important role for ADF/cofilin not only as a disassembly factor, but also as a potent F-actin network assembly factor through the creation of new barbed ends as in the model of dendritic nucleation within the lamellipodia of migrating cells (*DesMarais et al., 2004*; *Ghosh et al., 2004*; *Ichetovkin et al., 2002*). The combination of Fim1 and Adf1 generates dense F-actin bundles composed of many more actin filaments than Fim1 alone (*Figure 7A–B*) (*Skau et al., 2011*) It was previously shown that expression of an ADF/cofilin mutant deficient in severing results in delayed actin patch assembly in fission yeast, as a lack of severing prevents the creation of new mother filaments for actin patch initiation (*Chen and Pollard, 2013*). Our findings also potentially implicate Adf1 involvement in increasing the number of barbed ends within the dense actin patch network. The creation of barbed ends via severing takes advantage of the inherent polarity of an actin filament, as the newly created barbed end at the severed site is automatically oriented in the same direction as the original barbed end. This type of mechanism could be ideal for force-generating networks such as those at endocytic actin patches or the lamellipodia of migrating cells (*Ghosh et al., 2004*). However, the proper balance of assembly vs. disassembly must be achieved for proper actin patch dynamics, and likely involves the concerted effort of many ABPs.

## Competition and ABP sorting

Finally, our work highlights that the collective efforts of multiple ABPs can enhance or modulate the effects of individual ABPs on F-actin. There are many examples of multiple ABPs working together to create F-actin networks of defined organization and dynamics, including those within *Listeria* comet tails (*Loisel et al., 1999*), the leading edge of motile cells (*Blanchoin et al., 2000*), contractile F-actin networks (*Ennomani et al., 2016*; *Reymann et al., 2012*), and disassembling actin patches (*Jansen et al., 2015*). We show here that a similar idea holds true for competition-mediated ABP segregation. Tropomyosin Cdc8 is rapidly displaced from F-actin networks due to the combined activities of fimbrin Fim1 and ADF/cofilin Adf1. Though Cdc8 alone is capable of preventing association of Adf1 with F-actin (*Figure 6*), Fim1 rapidly displaces Cdc8 from actin filaments (*Figure 5*), allowing Adf1-mediated severing to occur (*Figure 8*). Adf1-mediated severing feeds back on Fim1's F-actin bundling activity by creating small, severed filaments that are easily incorporated into bundles (*Figure 7*), facilitating further bundling by Fim1 and increased Cdc8 displacement (*Figure 8*). Our work has implications not only for ABP sorting, but for the formation of any organized F-actin network that contains multiple components competing for the same binding substrate. Our lab has previously shown that competition for F-actin monomer is important for regulating the density and size of F-actin networks (*Burke et al., 2014*; *Suarez et al., 2015*). In addition, competitive interactions between DNA methylation and surrounding transcription factors mediate transcription factor association with certain regions of the genome (*Domcke et al., 2015*), and competition for membrane or receptor binding has been suggested to be involved in cargo sorting in a number of contexts (*Soza et al., 2004*). Overall, our work and the work of others suggest that competitive interactions between individual components can have large-scale effects on cellular organization in many contexts.

# Materials and methods

## Tropomyosin Cdc8 mutagenesis

All plasmid design and construction were performed using SnapGene software (from GSL Biotech; available at snapgene.com). Three amino acids in *S. pombe* tropomyosin Cdc8—Leucine 38, Isoleucine 76, and Aspartate 142—were chosen as potential labeling sites for mutation to cysteine based on four criteria: (1) localization on the outside of the coiled coil (at *b, c,* or *f* locations), (2) low sequence conservation amongst fungal tropomyosins (*Cranz-Mileva et al., 2013*), (3) present outside the first half of each period and therefore unlikely to affect Cdc8's association with F-actin (*Barua et al., 2013*), and (4) present away from C-terminal end so as to not affect end-to-end associations between Cdc8 molecules. To create Cdc8 mutants L38, I76C, and D142C for protein expression, QuikChange Site-Directed Mutagenesis (Agilent Technologies, Inc., Santa Clara, CA) was used to engineer distinct base pair substitutions within acetylation-mimic Cdc8 expression vector pET3a-AS-Cdc8 (*Monteiro et al., 1994*), (*Clayton et al., 2014*) and modifications were confirmed by sequencing. In vitro high-speed sedimentation assays and preliminary TIRFM assays at high Cdc8 concentrations determined that the I76C and D142C Cdc8 mutants behaved closest to wild-type Cdc8 in ability to bind to F-actin (*Figure 1—figure supplement 1B–D*). In addition, replacing the endogenous fission yeast *cdc8* gene with each of these mutants revealed that the I76C and D142C mutant strains behaved more similarly to wild-type than the L38C mutant (*Figure 1—figure supplement 2*). Therefore, a labeled I76C mutant was used in TIRFM assays for the rest of the study.

All three mutations were introduced within exon 2 of the *cdc8* gene. For insertion of *cdc8* mutants into the *S. pombe* genome, the portion of the *cdc8* mutant corresponding to exon 2 of *cdc8* was amplified from pET3a-AS-Cdc8*mut* protein expression vectors (using primers AAGGCGCGCCAGATCTAAAATTAATGCCGCTCGTGCTGAG and CAAGCTAAACAGATCTC TACAAATCCTCAAGAGCTTGGTGAAC), and the product was cloned into gene targeting vector pFA6-kanMX6 at BglII using In-Fusion HD Cloning (Clontech Laboratories, Mountain View, CA). cdc8*mut*-kan was then amplified (using primers AGGTATGAGATGATAGCTTTTCATTGGAAAA TCAAGTTGCTAATATTTGCTTTTTATTTAGAAAATTAATGCCGCTCGTGC and AGAAGATA TAAAAAGGTGGTATGTTTCTTCTATGTTCGTCAAGCTTTTCGCTATGAATTCGAGCTCGTTTAAAC) and transformed into a temperature sensitive mutant *cdc8-27* strain. Colonies were screened for absence of temperature sensitivity and resistance to kanamycin. Candidate colonies were then

screened for proper insertion by PCR and sequenced to confirm insertion at the *cdc8* locus. Strains created are listed in *Table 1*.

## ADF/Cofilin Adf1 mutagenesis

Three mutations were made in *S. pombe* ADF/cofilin Adf1 for labeling in TIRFM experiments. Endogenous cysteine residues were converted to alanine (C12S and C62A), and aspartate 34 was converted to cysteine (D34C) as previously described for ADF/cofilin from *S. cerevisiae* (*Suarez et al., 2011*). These mutations were made by QuikChange Site-Directed Mutagenesis in expression vector pMW-SpCofilin (*Skau and Kovar, 2010*). Modifications were confirmed by sequencing.

## Protein purification and labeling

Fimbrin Fim1, ADF/cofilin Adf1 (WT and mutant D34C, C12S, C62A), and tropomyosin AlaSer-Cdc8 (WT and L38C, I76C and D142C mutants) were expressed in BL21-Codon Plus (DE3)-RP (Agilent Technologies, Santa Clara, CA) and purified as described previously (*Skau and Kovar, 2010*). Briefly, Fim1 was purified with Talon Metal Affinity Resin (Clontech, Mountain View, CA). Adf1 was purified by an ammonium sulfate precipitation, size exclusion chromatography, and anion exchange chromatography. Cdc8 was purified by boiling the cell lysate, ammonium sulfate precipitation, and anion exchange chromatography. Actin was purified from chicken skeletal muscle or rabbit skeletal muscle acetone powder (Pel-Freez, Rogers, AR) as described previously (*Spudich and Watt, 1971*).

$A_{280}$ of purified proteins was taken using a Nanodrop 2000c Spectrophotometer (Thermo-Scientific, Waltham, MA). Protein concentration was calculated using extinction coefficients Fim1: 55,140 $M^{-1}$ $cm^{-1}$, Cdc8 (WT and mutants): 2980 $M^{-1}$ $cm^{-1}$, Adf1 (and mutant): 13,075 $M^{-1}$ $cm^{-1}$. Proteins were labeled with CF$^{TM}$405M (Sigma-Aldrich, St. Louis, MO), TMR-6-maleimide (Life Technologies, Grand Island, NY) or Cy5-monomaleimide (GE Healthcare, Little Chalfont, UK) dyes as per manufacturer's protocols immediately following purification, and were flash-frozen in liquid nitrogen and kept at −80°C. Cdc8 was reduced with DTT prior to labeling. For proteins labeled on one cysteine residue (Cdc8 and Adf1 mutants), labeling efficiency was determined by taking the absorbance at the emission max of the dye and calculating the coupling efficiency (*Kim et al., 2008*) . All reported Cdc8 concentrations are of the two-chain (dimer) molecule.

## Glass preparation for TIRFM

Coverslips and microscope slides (#1.5; Fisher Scientific) for TIRFM were prepared by washes in acetone, isopropanol, and water followed by sonication for 30 min in isopropanol. Washed glass was then cleaned by plasma cleaning for 3 min using a Harrick PDC-32G plasma cleaner (Harrick Plasma, Ithaca, NY). Cleaned coverslips and microscope slides were immediately passivated by incubation in 1 mg/mL PEG-Si (5000 MW) in 95% ethanol for 18 hr (*Winkelman et al., 2014*). Coverslips and slides were then rinsed in ethanol and water, and flow chambers were assembled as described previously (*Zimmermann et al., 2016*).

**Table 1.** Fission yeast strains used in this study

| Strain name | Genotype | Reference |
| --- | --- | --- |
| FY527 | h-, leu1-32, his3-D1, ura4-D18, ade6-M216 | Forsburg lab |
| MBY6663 | h+, pAct1 Lifeact-GFP::Leu+; ade6-m216; leu1-32; ura4-D18 | *Huang et al. (2012)* |
| KV920 | h?, cdc8-D142C::KanMX6, ade6-m216; ura4-D18 | This study |
| KV921 | h?, cdc8-I76C::KanMX6, ade6-m216; ura4-D18 | This study |
| KV922 | h?, cdc8-L38C::KanMX6, ade6-m216; ura4-D18 | This study |
| KV969 | h? cdc8-I76C::KanMX6, pAct1 Lifeact-GFP::Leu+, ade6-m216; ura4-D18 | This study |
| KV970 | h? cdc8-D142C::KanMX6, pAct1 Lifeact-GFP::Leu+, ade6-m216; ura4-D18 | This study |
| KV971 | h? cdc8-L38C::KanMX6, pAct1 Lifeact-GFP::Leu+, ade6-m216; ura4-D18 | This study |

## TIRFM

Time-lapse TIRFM movies were obtained using a cellTIRF 4Line system (Olympus) fitted to an Olympus IX-71 microscope with through-the-objective TIRF illumination and a iXon EMCCD camera (Andor Technology, Belfast, UK). Mg-ATP-actin (15% Alexa 488-labeled) was mixed with labeled or unlabeled ABPs and a polymerization mix (10 mM imidazole (pH 7.0), 50 mM KCl, 1 mM MgCl$_2$, 1 mM EGTA, 50 mM DTT, 0.2 mM ATP, 50 µM CaCl$_2$, 15 mM glucose, 20 µg/mL catalase, 100 µg/mL glucose oxidase, and 0.5% (400 centipoise) methylcellulose) to induce F-actin assembly (*Winkelman et al., 2014*). The mixture was then added to a flow chamber and imaged at 2.5 or 5 s intervals at room temperature.

## Fluorescence intensity line scans

Line scans were performed by drawing a three pixel width line along the actin filament and recording the fluorescence intensity along the line using 'Plot Profile' in FIJI (*Schindelin et al., 2012*; *Schneider et al., 2012*). The same region of interest was then applied to the ABP channel of interest. If necessary, the region of interest was adjusted to account for filament movement during channel switching.

## Tropomyosin Cdc8 coating of actin filaments

The occupancy of Cdc8 on actin filaments was determined from one frame from each TIRFM movie. For each movie, the frame of interest was chosen based on F-actin density (between 7 and 11 µm of filament per square µm) rather than time point. Segmented line ROIs were used to measure total actin filament length (488 channel) and total Cdc8 cable length (647 channel) at that frame. History of the TIRFM movie as well as Cdc8 fluorescence intensity was used to determine sites at which two Cdc8 cables were present on a single actin filament stretch, and in these cases each Cdc8 stretch was counted as a separate measurement. As there are two Cdc8 cables on a single actin filament, Total Cdc8 Occupancy=Total Cdc8 Length/(Actin Filament Length*2). Single vs double Cdc8 occupancy was determined by measuring the length of single- vs double-Cdc8-coated stretches and calculating the total length of single- or double-coated stretches divided by total actin filament length.

Free Cdc8 was calculated by ([Cdc8 added]−250*(Total Cdc8 Occupancy)/4)*0.001 (adapted from [*Hsiao et al., 2015*]). 250 (nM) refers to the F-actin concentration at the average time point that total Cdc8 occupancy was measured. The F-actin concentration was determined by a spontaneous pyrene actin assembly assay. Four refers to the ratio of bound Cdc8 to F-actin (1:4). The data were fit to a Hill equation $\theta=[L]^n/(K_d+[L]^n)$ where $\theta$ is the fraction of actin sites that are bound by Cdc8, [$L$] is the free Cdc8 concentration, $K_d$ is the apparent dissociation constant, and n is the Hill coefficient.

## Actin binding protein fluorescence intensity on actin filaments

The fluorescence intensity of fimbrin Fim1, tropomyosin Cdc8, or ADF/cofilin Adf1 was used to determine amount of ABP associated with actin filaments under different conditions. Analysis was performed on movie frames with a similar filament density for each compared condition. Segmented line ROIs (line width five pixels) were used to define each actin filament in the actin channel (488 channel). ROIs were then transferred to the ABP channel of interest and mean fluorescence intensity was measured for each actin filament. For comparing single actin filaments to F-actin bundles, the history of the actin channel and the actin fluorescence intensity was used to determine single filament versus bundled regions and separate ROI sets were generated and used to measure fluorescence intensity.

## Residence time measurements

To calculate residence times for individual ADF/cofilin Adf1 or tropomyosin Cdc8 molecules, a high total concentration of Adf1 or Cdc8 was included in the TIRFM assay to ensure total coating of the actin filaments (5 µM Adf1 or 2.5 µM Cdc8).~20% of Adf1 or Cdc8 was labeled with Cy5 dye in order to visualize the extent of coating of the protein on F-actin. The actin and Cy5-labeled Adf1 or Cdc8 was visualized every 10 s. A low (0.5–1%) percentage of Adf1 or Cdc8 was labeled with TMR in order to visualize single molecules, and fast imaging (five frames/sec) was performed in this channel. To measure residence time of Adf1 in the presence of Cdc8, 2.5 µM of unlabeled Cdc8 was included in

the reaction. Single molecules were tracked using MTrackJ (*Meijering et al., 2012*). Only single molecules that moved were tracked, as static molecules were assumed to be adsorbed to the coverglass. Both censored and uncensored events were obtained and a Kaplan-Meier analysis was performed (*Kaplan and Meier, 1958*). Events were fit to a single exponential $f(x) = f_0 e^{-\frac{x}{T_1}}$ that was used to determine residence time and $k_{off}$.

## Site of initial tropomyosin Cdc8 binding event

Site of the first tropomyosin Cdc8 binding event on an actin filament was determined by observing TIRFM movies performed at Cdc8 concentrations at the inflection point of the Hill plot (1.25 μM Cdc8). As our resolution limit is 100 nm, and individual Cdc8 molecules may only briefly associate with the actin filament before dissociating or forming a 'seed', we cannot determine explicitly the number of Cdc8 molecules in each of these initial association events. At the point of first observation of Cdc8 binding, the length of the actin filament was measured as well as the distance from the pointed end to the site of Cdc8 binding. The barbed and pointed ends of the actin filament were identified by observing photobleaching of the older, pointed end of the actin filament that occurred over time.

To compute the first binding times of a molecule to a substrate, we modeled the reaction as a master equation with two states, bound and unbound, with unbound transforming to bound at rate $k_{12}$ and the reverse process occurring with rate $k_{21} = 0$. Once a binding event is observed for a filament, that filament was unable to go back to having been never bound. We described P(t) as a vector representing the populations of the two states.

$$\frac{d\vec{P}}{dt} = \mathbb{W}\vec{P} \tag{1}$$

where W is a rate matrix whose columns must sum to zero. Hence:

$$\mathbb{W} = \begin{bmatrix} k_{22} & k_{12} \\ k_{21} & k_{11} \end{bmatrix} = \begin{bmatrix} 0 & k_{12} \\ 0 & -k_{12} \end{bmatrix} \tag{2}$$

In the case of binding to an extending F-actin substrate, if we assumed an average constant growth rate $v_{grow}$, the length of the substrate $l(t) = v_{grow}t$. With the assumption of uniform binding affinity, the rate of going from unbound to singly bound $k_{12}$ depended on time, therefore $k_{on}l(t) = k_{on}v_{grow}t$. $k_{on}$ is the rate per unit length of observing Cdc8 stably residing on a filament, which arises from the complex cooperative binding process discussed above and below.

*Equation 1* is solved formally by the equation

$$\vec{P}(t) = e^{\int_0^t \mathbb{W}(t')dt'} \vec{P}(0) \tag{3}$$

in which case:

$$\vec{P}(t) = \exp\left(\begin{bmatrix} 0 & \frac{k_{on}v_{grow}t^2}{2} \\ 0 & \frac{-k_{on}v_{grow}t^2}{2} \end{bmatrix}\right)\vec{P}(0) = \begin{bmatrix} 1 & 1 - e^{-\frac{k_{on}v_{grow}t^2}{2}} \\ 0 & e^{-\frac{k_{on}v_{grow}t^2}{2}} \end{bmatrix}\vec{P}(0) \tag{4}$$

Hence, since $\vec{P}(0) = \begin{pmatrix} P_2(0) \\ P_1(0) \end{pmatrix} = \begin{pmatrix} 0 \\ 1 \end{pmatrix}$, $P_1(t) = e^{-\frac{k_{on}v_{grow}t^2}{2}}$, the probability of a first binding event happening at time t = τ is the probability of still being unbound at time τ, $P_1(\tau)$, times the rate of binding at time τ, $k_{12}(\tau)$, hence:

$$P(\tau) = k_{12}(\tau)P_1(\tau) = k_{on}v_{grow}\tau \, e^{-\frac{k_{on}v_{grow}\tau^2}{2}} \tag{5}$$

Given this equation for the binding time, we also determined the probability of binding to a particular site on the actin filament that has been in the filament for an amount of time $\tau_{age}$, $P(\tau_{age})$.

Under our assumptions, at time t, the actin filament has length $l(t) = v_{grow}t$ and the probability of Cdc8 binding anywhere along the filament is uniform. The probability of binding at a distance $x_0$ from the end of the actin filament of length l is

$$P_{bind}(x_0|l) = \begin{cases} 1/l & l \geq x_0 \\ 0 & l_0 \end{cases} \qquad (6)$$

Consequently, given our assumption of a constant average growth rate $v_{grow}$, the probability of binding to a spot of a given age $\tau_{age}$ is

$$P_{bind}(\tau_{age}|l = v_{grow}t) = \begin{cases} \frac{1}{v_{grow}t} & \tau_{age} \leq t \\ 0 & \tau_{age} > t \end{cases} \qquad (7)$$

To determine the probability that a molecule binds to a part of the substrate of age $\tau_{age}$, we integrated *Equation 7* against the probability that the binding event happened at time t, as given by *Equation 5*. Therefore,

$$P_{bind}(\tau_{age}) = \int\limits_0^\infty dt\, P_{bind}(\tau_{age}|l = v_{grow}t)P(t) \qquad (8)$$

$$= \int\limits_{\tau_{age}}^\infty dt\, k_{on}e^{-\frac{k_{on}v_{grow}t^2}{2}} \qquad (9)$$

$$= \sqrt{\frac{k_{on}v_{grow}}{2}} \int\limits_{\tau_{age}\sqrt{\frac{k_{on}v_{grow}}{2}}}^\infty dx\, e^{-x^2} \qquad (10)$$

$$= \sqrt{\frac{\pi k_{on}v_{grow}}{2}}\mathrm{erfc}\left(\tau_{age}\sqrt{\frac{k_{on}v_{grow}}{2}}\right) \qquad (11)$$

where erfc(x) is the so-called complementary error function and was evaluated numerically. Using these equations, we chose $k_{on}$ to fit the data in *Figure 3—figure supplement 1* and we found that a value of $4 \times 10^{-6}$ sec$^{-1}$ nm$^{-1}$ gave good agreement with the data.

## Numerical simulation

The validity of the aforementioned expressions was tested by comparison with a simple simulation. The simulation methodology moreover provided a way to mimic the noise level that arose due to sample size limitations. A Monte Carlo scheme was performed by taking discrete time steps of size dt and at each time assuming the probability of binding in that time dt is

$$P_{bind}(dt) = 1 - e^{-k_{on}v_{grow}t\,dt} \qquad (12)$$

An example implementation:

```
from math import exp
from random import random
def simulate_binding(k_on, v_grow, dt):
    t = 0
    while True:
        fil_length = v_grow*t
        p_bind = 1 - exp(-k_on * v_grow * t * dt)
        # pick a random number
        #    from zero to one and decide
        #    if binding will occur
        if random()<p_bind:
            # choose a random position
            #    to bind from barbed end
```

```
        #   to pointed end
        binding_position = random()*fil_length
        binding_age = binding_position/v_grow
        return t,binding_age
    t = t + dt
def simulate_experiment(n_trials, k_on, v_grow, dt):
    binding_time_list = []
    age_list = []
    for i in range(n_trials):
        t,binding_age = simulate_binding(k_on,v_grow, dt)
        binding_time_list.append(t)
        age_list.append(binding_age)
    # histogram/bootstrap age_list
    # and binding_time_list
    ...
```

## Tropomyosin Cdc8 spreading rates

Elongation rates of individual tropomyosin Cdc8 cables were determined by creating a region of interest (ROI) of an individual actin filament in the actin 488 channel over many time points. The actin ROIs were then applied to the Cdc8 (647) channel and adjusted slightly if necessary to account for movement of the actin filament during channel switching. A kymograph of the Cdc8 647 channel was created from the ROIs using an adapted version of Kymograph - Time Space Plot ImageJ plugin (http://www.embl.de/eamnet/html/kymograph.html). Spreading rates were determined from kymographs by identifying examples of constant growth over at least 15 s (three frames). The distance of growth over time was calculated to determine a rate of spreading.

## Site of second tropomyosin Cdc8 binding event

To determine whether the two tropomyosin Cdc8 cables on each side of the actin filament are influenced by each other, we first quantified whether a second Cdc8 binding event on an actin filament was more or less likely to bind at a site that is already bound by Cdc8 on one side. First Cdc8 binding events were determined as stated above, and those actin filaments were observed until a second Cdc8 binding event occurred, and whether or not the binding occurred at a site already occupied by Cdc8 or not was determined. At that frame, the length of the actin filament as well as the total Cdc8 cable length was calculated to determine the current percent occupancy of Cdc8 on the actin filament. These 'current occupancies' were then binned into 0–12.5%, 12.5–25% or 25–50% occupancy. Initial occupancy cannot exceed 50% as one Cdc8 binding event could at most cover one side of the actin filament, or half of the available binding sites.

## Second binding probability analysis

To determine whether the proportion of observed second binding events that bind opposite a tropomyosin Cdc8-bound segment (*Figure 3E*) would be expected in the absence of indirect cooperativity, or whether they are more likely given inclusion of some positive or negative indirect cooperativity factor, we performed a bootstrapping-type analysis on the n = 37 observed events. For this analysis we asked, given experimental data, what is the probability that the second binding events are binding randomly vs binding in a biased fashion towards or against being across from a Cdc8-coated segment? For an actin filament of length $L_a$ and a first Cdc8 cable of length $L_1$ starting at position $x_1$ (in the range 0 to $L_a-L_1$) along the actin filament, we determined the probability of a second Cdc8 cable of length $L_2$ binding across from the first Cdc8 cable. We then compared the probability generated from different models with our experimental results. To account for experimental resolution, we divided the actin filament into a grid of 100 nm segments. In order to compute the probability of the second binding event overlapping with the first Cdc8 cable, we counted the number of places the second cable could bind that overlaps with the first cable of length $L_1$, and divided by the total number of potential places a second Cdc8 cable of length $L_2$ could bind. As a helical actin filament has two grooves, there are twice as many potential binding sites on F-actin

unoccupied by Cdc8 as those occupied by a single Cdc8 cable. However, if there is any overlap of second Cdc8 cable binding with the first Cdc8 cable, we assumed that there was only one potential binding face. For random binding, Cdc8 binding to each actin site was given an equivalent likelihood. To account for potential indirect cooperativity, we performed a weighted sum, where the second Cdc8 cable binding is proportionally more or less likely by a factor of $c$ at all sites overlapping with the first cable of Cdc8. The probability of an overlapping binding was therefore given by the following three expressions:

$$N_1 = 2 \sum_{x=0}^{L_a-L_2} ((x+L_2)<x_1)|(x>(x_1+L_1))$$

$$N_2 = \sum_{x=0}^{L_a-L_2} \begin{cases} (x_1-x)+c(L_2-(x_1-x)) & x<x_1 \ \& \ (x+L_2)>x_1 \\ (L_2-((x_1+L_1)-x))+c((x_1+L_1)-x) & x \geq x_1 \ \& \ (x<x_1+L_1) \end{cases}$$

(with the terms in outer parentheses being restricted to minimum value 0 and maximum $L_2$)

$$p_2 = \frac{N_2}{N_1+N_2}$$

We then compared the expected outcomes for random ($c = 1X$) vs positive or negative indirect cooperativity values ($c = 2X$ or $c = 0.5$ respectively) (*Figure 3F*). Finally, we performed a bootstrapping analysis. For each experimentally observed set of events, where a given ($L_a$, $L_1$, and $x_1$) was measured, we computed the probability that a cable of length $L_2$ binding would overlap with the first Cdc8 cable ($p_2$). We then re-performed the 37 experiments 5000 times with a chosen indirect cooperativity factor $c$, choosing randomly with probability $p_2$ whether or not overlapping binding occurred. Binning these events similar to the experimental data, we computed the values shown in *Figure 3F*.

## Lattice model of tropomyosin Cdc8 loading and spreading

To probe the microscopic origins of the high cooperativity observed for tropomyosin Cdc8 (*Figure 1B*), we distinguished between two potential cooperative mechanisms: (1) end-to-end cooperativity and (2) indirect cooperativity (*Figure 1C*). We distinguished between these two models by modeling the binding kinetics of Cdc8 to a growing actin filament using a lattice model (*Figure 4Bi*, 4 Ci). As two distinct Cdc8 cables can bind to an actin filament (one on the surface of each groove of the helical actin filament), we represented the actin filament as a lattice with two rows representing the two actin surfaces potentially bound by Cdc8. Each Cdc8 molecule interacts across four actin monomers, meaning that a bound Cdc8 extends over the length of 8 actin monomers total. Therefore, as each Cdc8 cable is represented separately, we ascribed to each lattice site a length $l$ of approximately 21.6 nm (assuming an approximate actin spacing of 2.7 nm). The dynamics of Cdc8 molecules associating with this lattice were then simulated using a kinetic Monte Carlo procedure (*Newman and Barkema, 1999*).

First, we considered the case for a non-elongating actin filament, represented by a fixed lattice of length N. Each potential Cdc8 binding site can either be occupied (1) or unoccupied (0) by Cdc8. We described binding site site $i$ in row $j$ as $x_{ij}$. Every unoccupied site had a base on-rate $k_{on}$ (0→1) and a base off-rate $k_{off}$ (1→0). To include the effect of end-to-end cooperativity, the on-rate for unoccupied site $ij$ was multiplied by a factor of w and the off-rate for occupied site $ij$ was divided by a factor of w for each occupied immediate neighbor in the same row (*Figure 4Bi*). Using these parameters, we ran a model describing Cdc8 binding with end-to-end cooperativity as the sole form of cooperativity (*Figure 4B*). In order to additionally probe the potential role of indirect cooperativity, we adjusted the same model to include additional parameters for indirect cooperativity. In this model, on-rates were multiplied by a factor c and off-rates were divided by a factor c for each occupied site in the opposite row (row 1-j) (*Figure 4Ci*). For a lattice size 2xN sites, there were precisely 2xN possible events. The dynamics were then solved using kinetic Monte Carlo simulation (*Newman and Barkema, 1999*) up to a chosen time *max_time* or until the lattice was completely filled. In order to factor in an elongating actin filament, an additional type of event representing actin filament growth was included, with a rate equal to the input growth rate of actin, $v_{grow}$. In this

case, whenever that event was selected by the algorithm, time was advanced and N was set equal to

N = floor(new_time*site_extension_rate)

To make kymographs and to compute the occupancy of Cdc8 as accurately as possible, we processed the output raw simulation data in three ways:

1. In TIRFM experiments, ~20% of Cdc8 molecules were labeled. Hence, in our kymographs, we replaced 80% of the occupied sites with a value of 0.
2. The experiments were limited by optical resolution. To approximate this, each remaining occupied site was broadened by filling in adjacent sites in the same row within a radius of 100 nm (±4 sites), the experimental resolution.
3. The two rows were summed together to give fluorescence values of 0, 1, or two at each site along the actin filament.

The coverage data in *Figure 4Biii and 4Ciii* as computed by averaging this fluorescence level over the filament at the time when the filament had length 6 μm over 24 such simulations. 6 μm was the average length of the actin filaments at the frame used for quantification of Cdc8 coverage in TIRFM experiments (4Aiii). In order to choose the values of w, c and $k_{off}$, we first fixed c, and tried combinations of $k_{off}$ and w that gave (a) the best match to the data shown in *Figure 1B* as $k_{on}$ is scanned, and (b) gave kymographs whose first time for observed binding was similar to what was observed experimentally. Due to cooperative binding effects, modeling is required to find a value for $k_{on}$ that is commensurate with experimental data. In *Figure 4—figure supplement 1*, $k_{off}$ was measured from single molecule Cdc8 events on an actin filament fully coated by Cdc8. In our model, the microscopic $k_{off}$ corresponding to this situation was $k_{off}^{measured} = k_{off}^{model}/w^2/c$. Using $k_{off}^{model}$ values far from those measured in *Figure 4—figure supplement 1* did not give a good agreement with the data. Constraining $k_{off}$ to the value measured experimentally results in a single value of w that best matched the experimental data (*Figure 1B*, parameters $k_{off}$ = 116.8 sec$^{-1}$, w = 40, c = 1, $K_d$ = 0.08 sec$^{-1}$), with the results from these simulations shown in *Figure 4B*. However, these simulations did not match the experimental single/double coverage data (*Figure 4Aiii*). Relaxing the restraint on $k_{off}$ did not alleviate this problem. For a fixed value of c > 1, there is a single w that best matched the experimental data. For c = 1.25, while these simulations matched the experimental data (*Figure 1B*), they over-stabilized double cable association. Hence, we slightly relaxed the restriction on $k_{off}$ and the simulations with parameters ($k_{off}$ = 300 sec$^{-1}$, w = 125, c = 1.25, $K_d$ = 0.02 sec$^{-1}$) matched the experimental data (*Figure 4A*) very closely as well as qualitatively reproducing many of the aspects we observed for Cdc8 loading (*Figure 4C*).

We note that a limitation of this model is that it did not allow for any 'frame shifts' in the Cdc8 binding. Hence there cannot be defects in Cdc8 occupancy on a cable that are smaller than four actin monomers in these representations, and we did not allow the Cdc8 to 'slide' except by binding and unbinding. The first part of this could be addressed by using a lattice of 4x higher resolution. However, since we did not know how to account for the end-to-end cooperativity in this case, or how to represent the sliding dynamics, we chose the simpler representation for the purposes of this study.

The rates of a lattice with these w and c parameters fixed can be written in terms of energies for a two-row Ising model in a field. Since this system is still one dimensional in nature, for the infinite-length case any equilibrium properties of the system can be derived using a transfer matrix approach (*Tsuchiya and Szabo, 1982*) in the same way as for a standard Ising model (except that the transfer matrix is 4 × 4). However, given that the growth of the actin filament and the way that the data is analyzed play significant roles in the coverage fractions computed, we have found the simulations useful since they are able to replicate the full dynamics as shown in *Figure 4Bii and 4Cii*.

## Bundling quantification

The percentage of actin filaments bundled was quantified at similar F-actin densities (between 800 and 1100 μm per field) for each experiment. The total actin filament length in the chamber was measured manually by creating ROIs for every actin filament. ROIs for every segment of actin filament present in a bundle were then created, and the ratio of actin filament present in a bundle vs. total actin filament length was quantified.

## Quantification of fold-change in fluorescence intensity

Fold-change in fluorescence intensity over time was taken by taking a total fluorescence measurement of the entire TIRFM field for each frame of the movie. The fold-change in fluorescence of each frame compared to the first frame of the movie was quantified and plotted over time.

## Severing rate quantification

ROIs were created for all single filaments and two-filament bundles at that time point and total filament length was measured. Severing events that occurred within those filaments prior to the selected frame were quantified, and severing events per micron per second was calculated. We were only able to directly observe severing of single filaments and two-filament bundles, and were unable to determine severing rate within the dense bundles generated by a combination of fimbrin Fim1 and ADF/cofilin Adf1.

## Pyrene assembly assays

The spontaneous assembly of 1.5 µM Mg-ATP-actin monomers (10% pyrene-labeled) was carried out in a 96 well plate as described (*Zimmermann et al., 2016*). WT ADF/cofilin Adf1 or TMR labeled Adf1 were mixed with 10X KMEI (500 mM KCl, 10 mM MgCl$_2$, 10 mM ethylene glycol tetraacetic acid [EGTA], and 100 mM imidazole, pH 7.0) and placed in the lower row of the plate. Pyrene fluorescence readout over time was detected using an Infinite M200 Pro (Tecan Systems, Inc., San Jose, CA) fluorescence plate reader.

## DAPI/Calcofluor

Fission yeast cells were grown in YE5S media at 25°C for 24–36 hr. Cells were fixed in 100% cold methanol and nuclei (DAPI) and septa (calcofluor) were visualized. For staining, cells were incubated in 300 µL of 50 mM sodium citrate and 4 µL Calcofluor White Stain (Fluka Analytical, Sigma-Aldrich, St. Louis, MO) at 37°C for 5 min. Cells were then washed with 1 mL of 50 mM sodium citrate, and resuspended in 15 µL sodium citrate and 4 µL DAPI stock (1 mg/mL in H20, Life Technologies, Carlsbad, CA). Stained cells were kept on ice until imaging. Cells were imaged using differential interference contrast (DIC) and epifluorescence with an Orca-ER camera (Hamamatsu, Bridgewater, NJ) fitted to an IX-81 microscope (Olympus, Tokyo, Japan), with a 60 × 1.4 N.A. Plan Apo objective.

## Time of ring assembly

Time required for fission yeast strains to assemble a contractile ring was quantified using strains expressing Lifeact-GFP as a general actin marker (*Table 1*). Single z-plane movies were taken (one frame per minute) using an epifluorescence microscope as in DAPI/calcofluor imaging. Time was quantified from the first appearance of fluorescence at the midzone to the formation of a compact ring.

## Phallicidin staining

BODIPY-phallacidin staining of fission yeast to visualize their actin cytoskeleton was performed as described previously (*Sawin and Nurse, 1998*). BODIPY-phallacidin powder (Thermo Fisher Scientific, Waltham, MA) was suspended to 0.2 units/µL in methanol, aliquoted, and lyophilized for storage at −20°C. Fission yeast grown in YE5S were fixed in 16% paraformaldehyde at room temperature for 5 min. Cells were washed with PEM buffer three times at room temperature and permeabilized in PEM with 1% triton X-100 for exactly 1 min. Cells were then spun at 7000 RPM for 30 s and the supernatant was removed quickly. Cells were washed in PEM buffer three times and resuspended in 10 µL PEM buffer following the final wash. Lyophilized phallacidin was resuspended to one unit/µL and 1 µL of resuspended phallacidin was added to 10 µL of cells and incubated at room temperature for 30 min in the dark. Following phallacidin incubation, cells were washed with 1 mL PEM and spun at 7000 RPM for 30 s or until nearly all cells were spun down. Supernatant was removed leaving the resuspended cells in a small volume. Resuspended cells were imaged on glass slides using a Zeiss Axiovert 200M fitted with a 100x, 1.4 NA objective and Yokogawa CSU-10 spinning disk unit (McBain, Simi Valley, CA) equipped with a 50-milliwatt 473 nm DPS laser and Cascade 512B EM-CCD camera (Photometrics, Tucson, AZ).

## High-Speed sedimentation assays

Sedimentation assays were performed as previously described (*Skau and Kovar, 2010*). 15 µM Mg-ATP actin monomers were spontaneously assembled in 10 mM imidazole, pH 7.0, 50 mM KCl, 5 mM MgCl$_2$, 1 mM EGTA, 0.5 mM DTT, 0.2 mM ATP and 90 µM CaCl$_2$ for 1 hr to generate F-actin. F-actin was then incubated with tropomyosin Cdc8 mutants (2 µM) for 20 min at 25°C and spun at 100,000g at 25°C. Supernatant and pellets were separated by 15% SDS-PAGE gel electrophoresis and stained with Coomassie Blue for 30 min, destained for 16 hr and analyzed by densitometry with ImageJ.

## Acknowledgements

This work was supported by NIH R01 GM079265 and ACS RSG-11-126-01-CSM (to DRK), NIH MCB Training Grant T32 GM0071832 (to JRC and KEH), NSF Graduate Student Fellowship DGE-1144082 (to JRC), NIH Ruth L Kirschstein NRSA F32 GM113415-01 (to GMH) and NIH R01 GM093965 (to SEH-D). Additional support was provided to DRK and GAV by the University of Chicago MRSEC, funded by the NSF through grant DMR-1420709. We thank Ben Glick and GSL Biotech for the use of SnapGene for plasmid construction, and DJ Speed for assistance with cloning of Adf1 mutants. We also thank Suri Vaikuntanathan, the Kovar lab, and the Voth lab for helpful discussions.

## Additional information

### Funding

| Funder | Grant reference number | Author |
| --- | --- | --- |
| National Institutes of Health | GM079265 | David R Kovar |
| American Cancer Society | RSG-11-126-01-CSM | David R Kovar |
| National Science Foundation | DGE-1144082 | Jenna R Christensen |
| National Institutes of Health | T32 GM0071832 | Jenna R Christensen Kaitlin E Homa |
| National Institutes of Health | F32 GM113415-01 | Glen M Hocky |
| National Institutes of Health | GM093965 | Sarah E Hitchcock-DeGregori |
| National Science Foundation | DMR-1420709 | Gregory A Voth David R Kovar |

The funders had no role in study design, data collection and interpretation, or the decision to submit the work for publication.

### Author contributions

JRC, Conceptualization, Formal analysis, Funding acquisition, Investigation, Writing—original draft, Writing—review and editing; GMH, Formal analysis, Funding acquisition, Investigation, Writing—review and editing; KEH, ANM, Formal analysis, Investigation; SEH-D, Supervision, Funding acquisition, Methodology, Writing—review and editing; GAV, Supervision, Funding acquisition, Writing—review and editing; DRK, Conceptualization, Supervision, Funding acquisition, Writing—review and editing

### Author ORCIDs

Jenna R Christensen, http://orcid.org/0000-0003-0323-6169
Glen M Hocky, http://orcid.org/0000-0002-5637-0698
David R Kovar, http://orcid.org/0000-0002-5747-0949

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
