## [Decision Letter]

Thank you for submitting your article "Competition between Tropomyosin, Fimbrin, and ADF/Cofilin drives their sorting to distinct actin filament networks" for consideration by *eLife*. Your article has been reviewed by three peer reviewers, one of whom, Mohan K Balasubramanian (Reviewer #1), is a member of our Board of Reviewing Editors, and the evaluation has been overseen by Anna Akhmanova as the Senior Editor. The following individuals involved in review of your submission have agreed to reveal their identity: Mithilesh Mishra (Reviewer #2); Fred Chang (Reviewer #3).

The reviewers have discussed the reviews with one another and the Reviewing Editor has drafted this decision to help you prepare a revised submission.

The editor and referees think that your work on characterizing the competition between tropomyosin, fimbrin, cofilin in actin network organization is exciting.

It appears that most of the referee comments can be fixed by rewriting and a better discussion. Please see a consolidated list of referee comments and our expectation during revision in brackets.

1) Is the competitive relationship between Fim1 and Cdc8 specific or do other bundling proteins compete with Cdc8 for F-actin binding? The authors have the tools and can easily address potential competition between Ain1 / Fim1/ Cdc8. This becomes particularly important if one considers some of the work of Doug Robinson and colleagues who have investigated micro and nanosecond occupancies of Fimbrin, A-actinin, and cortexilin on actin (Experiment or discussion).

2) East et al. (Biochem J) have described tropomyosin mutants that lack the ability for dimer formation. The authors could test use mutant protein as a control to demonstrate Cdc8 cooperativity. This paper was not referred and I hope the authors will rectify this (This is essential).

3) Adf1 preferentially binds to ADP-F-actin and Cdc8 has no preference for the ATP/ADP state of F-actin. How about Fim1? This can be informative on mechanisms of competition to load key binding proteins onto actin filaments. (Experiment or discussion).

4). Indirect Cooperativity of Tropomyosin:

The authors propose that Tropomyosin engages in both end to end binding and therefore direct cooperative along with an ill-defined indirect cooperativity that facilitates preferential loading of tropomyosin on the opposing actin surface. This is evidenced by an increased Fluorescence intensity of tropomyosin along actin filament. This mechanism of such cooperativity merits some discussion. Tropomyosin mediated actin stabilisation can, in principle, lead to increased probability of tropomyosin loading based on an increased lifetime of a stabilised actin filament. Also the initial loading event whether across or along the filament gives rise to a spike in intensity (taken as the seed of the loading event), that seems to become uniform over time. Does the seed lose its structure? Does it unload after initiating the reaction? Or is it an artefact of photo-bleaching? (discussion).

5) Figure 1—figure supplement 1:a) From text (Results section) it appears that the authors suggest that the Cdc8 (I76C) more closely resembles the wild type tropomyosin than either L38C or D142C. In the figures however, D142C seems strikingly identical to I76C, if not marginally better (1E, 1F: Quantification of cytokinesis abnormality). Therefore it is not clear to me why I76C was chosen over D142C. (explanation/discussion)b) Figure 1—figure supplement 1:

The Y-axis reads "Cdc8 pellet normalised to highest value"

It's not clear to the reader what highest value is it normalised to. Is it the highest cdc8 pellet recovered or the actin amount. Either case I think it's better to plot cdc8 (pellet) normalised to its associated actin amount. (figure modification).

6) Subsection “Tropomyosin Cdc8 is actively displaced from F-actin bundled by fim1.”a) The authors make a strong closing argument in this section –

"These findings suggest that the cooperativity of Cdc8 assists not only its assembly onto filaments but also its rapid displacement specifically from F-actin networks bundled by Fim1." However all the evidence strictly points to a fim1 mediated displacement of cdc8 and not any involvement of cdc8's cooperativity in this unloading. I guess that the rapidity of cdc8 stripping lead the authors to this conclusion. Essentially to claim this the authors need to show that fim1 bundling even in a restricted localised zone leads to rapid eviction of the entire tropomyosin filament. Otherwise and as it stands currently, fim1 coats the entire actin filament along its length and consequently cdc8 is excluded. The fast dynamics can be attributed to rapid actin bundling (fim1 association) and physical eviction of tropomyosin without invoking tropomyosin cooperativity. (perform an experiment or tone down the conclusions).b) The authors, although not explicitly, seem to wrongly indicate the contractile ring is devoid of cofillin. (clarify this point).

---

## [Author Response]

*The editor and referees think that your work on characterizing the competition between tropomyosin, fimbrin, cofilin in actin network organization is exciting.*

*It appears that most of the referee comments can be fixed by rewriting and a better discussion. Please see a consolidated list of referee comments and our expectation during revision in brackets.*

*1) Is the competitive relationship between Fim1 and Cdc8 specific or do other bundling proteins compete with Cdc8 for F-actin binding? The authors have the tools and can easily address potential competition between Ain1 / Fim1/ Cdc8. This becomes particularly important if one considers some of the work of Doug Robinson and colleagues who have investigated micro and nanosecond occupancies of Fimbrin, A-actinin, and cortexilin on actin (Experiment or discussion).*

We agree that this is an extremely interesting and important question, which is a major area of investigation that we have been actively pursuing. In fact, we are preparing a new manuscript describing this work that we will soon be submitting to *eLife* as an accompanying 'Research Advance' article that is tentatively entitled "Cooperation between Tropomyosin Cdc8 and α-actinin Ain1 prevents fimbrin Fim1 association with contractile ring F-actin." In brief, we have discovered that the competitive relationship is specific to Fim1 and Cdc8.

Conversely, (1) Ain1 and Cdc8 do not compete with each other, and instead (2) Cdc8 actually enhances Ain1-mediated bundling, which (3) allows Ain1 and Cdc8 to collectively compete effectively for actin filaments with Fim1. Our 'Research Advance" manuscript also focuses on the importance of occupancy time of these ABPs in their sorting to distinct F-actin networks in fission yeast.

We also agree that it is important to reference work from the Robinson lab and the importance of dynamics in mediating ABP sorting, and have added a new section to the Discussion: “We suspect that Fim1’s long residence time on F-actin bundles enhance its ability to create regions of high Fim1 occupancy that displace Cdc8, suggesting that ABP dynamics (on-/off-rate) may partially dictate the ability of an ABP to compete. An ABP’s dynamics could be driven by its intrinsic biochemical properties (Winkelman et al., 2016), post- translational modifications (Miao et al., 2016), or mechanical stresses within the F-actin network (Srivastaya and Robinson, 2015), resulting in the fine-tuning of ABP sorting by many factors.”

*2) East et al. (Biochem J) have described tropomyosin mutants that lack the ability for dimer formation. The authors could test use mutant protein as a control to demonstrate Cdc8 cooperativity. This paper was not referred and I hope the authors will rectify this (This is essential).*

The mutants referred to in this paper are still capable of forming Cdc8 dimers, but have mutations in the end-to-end binding regions that inhibit or enhance the ability of Cdc8 to polymerize into cables. Due to the unusual binding dynamics of Cdc8 on F-actin, it is difficult to assess our model of Cdc8 indirect cooperativity using these mutants. An ideal experiment would enhance the ability of Cdc8 to bind to F-actin while inhibiting its end-to-end binding. However, a mutant that is able to strongly associate with actin without being able to polymerize has not been identified. Therefore, though these mutants are certainly interesting, it would be difficult to address our model of indirect cooperativity with use of these mutants, as they only inhibit or enhance ability of Cdc8 to polymerize. We agree that this is an important paper to refer to and have included the following new statement: “Mutations in the N- or C-terminal domains of Cdc8 affect its ability to bind cooperatively and polymerize on an actin filament (East et al., 2011).”

*3) Adf1 preferentially binds to ADP-F-actin and Cdc8 has no preference for the ATP/ADP state of F-actin. How about Fim1? This can be informative on mechanisms of competition to load key binding proteins onto actin filaments. (Experiment or discussion).*

Our lab has previously determined that fimbrin Fim1 has no preference for the ATP/ADP state of F-actin (Skau et al., 2011). We agree that the mechanisms of initial loading of key ABPs is particularly interesting and is the subject of current research in our lab.

*4) Indirect Cooperativity of Tropomyosin:*

*The authors propose that Tropomyosin engages in both end to end binding and therefore direct cooperative along with an ill-defined indirect cooperativity that facilitates preferential loading of tropomyosin on the opposing actin surface. This is evidenced by an increased Fluorescence intensity of tropomyosin along actin filament. This mechanism of such cooperativity merits some discussion. Tropomyosin mediated actin stabilisation can, in principle, lead to increased probability of tropomyosin loading based on an increased lifetime of a stabilised actin filament. Also the initial loading event whether across or along the filament gives rise to a spike in intensity (taken as the seed of the loading event), that seems to become uniform over time. Does the seed lose its structure? Does it unload after initiating the reaction? Or is it an artefact of photo-bleaching? (discussion).*

We agree that the mechanism of indirect cooperativity merits further discussion, and have also thought that this may be a result of stabilization of the actin filament, particularly as it is known that Cdc8 enhances the persistence length of F-actin. We have added new additional discussion:

“Tropomyosin is known to increase the persistence length of F-actin (Fujime and Ishiwata, 1971; Isambert et al., 1995). As a result, the stabilizing effect of an initial Cdc8 binding to one side of the actin filament may favor binding of a second Cdc8 to the opposing side. Additionally, if the rigidity provided by initial Cdc8 binding is propagated slightly further up or down the actin filament, more tropomyosin molecules could associate with the actin filament as a result of an increase in F-actin rigidity.”

The uniformity over time in fluorescence appears to be an artifact of photobleaching, as it happens faster when we image more frequently. However, we cannot rule out the possibility that tropomyosin molecules are leaving the cable, potentially at ‘gap’ regions. The longer cables appear to remain intact, though we do observe small seeds (especially at lower concentrations of tropomyosin) that do grow and shrink more frequently.

*5) Figure 1—figure supplement 1:a) From text (Results section) it appears that the authors suggest that the Cdc8 (I76C) more closely resembles the wild type tropomyosin than either L38C or D142C. In the figures however, D142C seems strikingly identical to I76C, if not marginally better (1E, 1F: Quantification of cytokinesis abnormality). Therefore it is not clear to me why I76C was chosen over D142C. (explanation/discussion).*

The I76C and D142C Cdc8 mutants are both suitable for TIRF microscopy experiments, and in fact, we use both in the lab. However, we have found that I76C has been easier to label well consistently, for reasons that are unclear. Additionally, our updated analysis both in vitro and in vivo (Figure 1—figure supplement 1 and Figure 1—figure supplement 2) show that I76C and D142C are very similar and both are viable options for experiments. We have added new additional characterization that further informs our decision on which mutant to use: “We examined the functionality of each Cdc8 mutant in vitro and in vivo to identify the mutant most similar to wild-type Cdc8. Two mutants, Cdc8(I76C) and Cdc8(D142C), bound F-actin similarly to wild type Cdc8 (Figure 1—figure supplement 1), were able to be labeled and visualized by TIRFM (100% of actin filaments associated with tropomyosin, Figure 1—figure supplement 1), and caused only very mild cytokinesis defects as the sole copy of Cdc8 in fission yeast (Figure 1—figure supplement 2). Therefore, we chose one of these mutants, Cdc8(I76C) for further study in TIRFM experiments.”

*b) Figure 1—figure supplement 1:*

*The Y-axis reads "Cdc8 pellet normalised to highest value"*

*It's not clear to the reader what highest value is it normalised to. Is it the highest cdc8 pellet recovered or the actin amount. Either case I think it's better to plot cdc8 (pellet) normalised to its associated actin amount. (figure modification).*

These plots were normalized to the highest Cdc8 recovered. When we plot Cdc8 in the pellet normalized to its associated actin, all mutants appear to bind F-actin similarly. We have changed the graph to show Cdc8 pellet normalized to associated actin (Figure 1—figure supplement 1).

*6) Subsection “Tropomyosin Cdc8 is actively displaced from F-actin bundled by fim1.”a) The authors make a strong closing argument in this section –*

"These findings suggest that the cooperativity of Cdc8 assists not only its assembly onto filaments but also its rapid displacement specifically from F-actin networks bundled by Fim1." However all the evidence strictly points to a fim1 mediated displacement of cdc8 and not any involvement of cdc8's cooperativity in this unloading. I guess that the rapidity of cdc8 stripping lead the authors to this conclusion. Essentially to claim this the authors need to show that fim1 bundling even in a restricted localised zone leads to rapid eviction of the entire tropomyosin filament. Otherwise and as it stands currently, fim1 coats the entire actin filament along its length and consequently cdc8 is excluded. The fast dynamics can be attributed to rapid actin bundling (fim1 association) and physical eviction of tropomyosin without invoking tropomyosin cooperativity. (perform an experiment or tone down the conclusions)

We agree that Cdc8 displacement appears to be directly mediated by Fim1, but also posit that Cdc8 cooperativity may enhance its Fim1-mediated removal, resulting in an ‘all-or-nothing’ removal of Cdc8 cables from the actin filament across short ranges. The stripping of Cdc8 cables from initial sites of removal is what led us to this conclusion, but we agree that we have overstated the claim. We have toned down our conclusions and it now reads “These findings demonstrate that Fim1-mediated bundling displaces Cdc8 from the actin network. Additionally, as Cdc8 appears to be rapidly ‘stripped’ from a few initial dissociation points, it is possible that the cooperativity of Cdc8 assists not only its assembly onto filaments but also its rapid displacement specifically from F-actin networks bundled by Fim1.”

*b) The authors, although not explicitly, seem to wrongly indicate the contractile ring is devoid of cofillin. (clarify this point).*

We agree that the wording is confusing and have clarified the point to now read “Though Cdc8's presence at the contractile ring and actin cables likely prevents the unwanted association of several types of ABPs at those F-actin networks, opposing mechanisms must be in place to prevent Cdc8 from associating with actin patches“.